# Quantifying burning efficiency in Megacities using $NO_2$/CO ratio from the Tropospheric Monitoring Instrument (TROPOMI)

Srijana Lama[1], Sander Houweling[1,2], K. Folkert Boersma[3,4], Ilse Aben[2,5], Hugo A C Denier van der Gon[6], Maarten C. Krol[3,7], A.J.(Han) Dolman[1], Tobias Borsdorff [2], Alba Lorente[2]

[1]Vrije Universiteit, Department of Earth Sciences, Amsterdam, the Netherlands
[2]SRON Netherlands Institute for Space Research, Utrecht, the Netherlands
[3]Wageningen University, Meteorology and Air Quality Section, Wageningen, the Netherlands
[4]Royal Netherlands Meteorological Institute, R&D Satellite Observations, de Bilt, the Netherlands
[5]Vrije Universiteit, Department of Physics and Astronomy, Amsterdam, the Netherlands
[6]TNO, Department of Climate, Air and Sustainability, Princetonlaan, the Netherlands
[7]Institute for Marine and Atmospheric Research Utrecht, Utrecht University, Utrecht, the Netherlands

*Correspondence to*: Srijana Lama (s.lama@vu.nl and sreejanalama@gmail.com)

**Abstract**. This study investigates the use of co-located $NO_2$ and CO retrievals from the TROPOMI satellite to improve the quantification of burning efficiency and emission factors over the mega-cities of Tehran, Mexico City, Cairo, Riyadh, Lahore and Los Angeles. *NOx (NO+NO₂) emission increases during the efficient combustion whereas incomplete combustion results to higher CO emission. Therefore, $NO_2$/CO is a good proxy for combustion efficiency*. Local enhancements of CO and $NO_2$ above megacities are well captured by TROPOMI at relatively short averaging times. In this study, the Upwind Background and Plume rotation methods are used to investigate the accuracy of satellite derived $\Delta NO_2/\Delta CO$ ratios. The column enhancement ratios derived using these two methods vary by 5 to 20 % across the selected megacities. TROPOMI derived column enhancement ratios are compared with emission ratios from the EDGAR v4.3.2 and MACCity, 2018 emission inventories. TROPOMI correlates strongly (r = 0.85 and 0.7) with EDGAR and MACCity showing the highest emission ratio for Riyadh and lowest for Lahore. However, inventory derived emission ratios are higher by 60 to 80 % compared to TROPOMI column enhancement ratios across the six megacities. The short lifetime of $NO_2$ and different vertical sensitivity of TROPOMI $NO_2$ and CO explain most of this difference. We present a method to translate TROPOMI retrieved column enhancement ratios into corresponding emission ratio, accounting for these influences. Except for Los Angeles and Lahore, TROPOMI derived emission ratios are close (within 10 to 25%) to MACCity. For EDGAR, however, emission ratios are higher by ~65 % for Cairo, 35 % for Riyadh and ~70 % for Los Angeles. The air quality monitoring networks in Los Angeles and Mexico City are used to validate the use of TROPOMI. Over Mexico City, these measurements are consistent with TROPOMI, EDGAR and MACCity derived emission ratios. For Los Angeles, however, EDGAR and MACCity are higher by a factor 3 compared to TROPOMI. The ground-based measurements are consistent with a poorer burning efficiency in Los Angeles as inferred from TROPOMI, demonstrating its potential to monitor burning efficiency.

# 1    Introduction

The rapid urbanization and economic growth in developing countries has led to a strong increase in urban air pollution (Pommier et al., 2013; United Nations, 2018). In the south Asian cities of Kabul and Dhaka, for instance, nitrogen dioxide ($NO_2$) increases have been reported in the order of 10 % yr$^{-1}$ (Schneider et al., 2015). In New Delhi, emissions of carbon monoxide (CO) increased by 22.4 % in the period 2000-2008 (Jiang et al., 2017). In European countries, on the other hand, the use of modern technology and other air pollution abatement measures have decreased $NO_2$ concentrations by 10-50 % in
the period of 2004 to 2010 (Castellanos & Boersma, 2012) and CO by 35 % between 2002 to 2011 (Guerreiro et al., 2014). To develop effective air pollution control strategies, accurate information on local emission sources and combustion processes is important (Borsdorff et al, 2004). However, developing countries and remote areas lack the local infrastructure needed to obtain detailed information e.g. about energy consumption, fuel type and technology. Limited process information contributes largely to the uncertainty in emission inventories (Silva & Arellano, 2017). *For example, the range of*
*uncertainty in the Chinese NOx and CO emissions in the period of 2005 to 2008, has been estimated at - 20 to +45% due to inadequate information about the fuel consumption and rough estimates of emission factor (Zhao et al., 2011, 2012).* In the global emission inventory EDGAR v4.3.2, uncertainties in regional emissions have been estimated at 17 to 69% for NOx, and 25 to 64% for CO (Crippa et al., 2016). In this study, we investigate the use of satellite remote sensing to improve the emission quantification for these important air pollutants.

In global emission inventories, combustion related emissions are computed as the product of the amount of fuel burned (activity data), and the composition of the emissions as represented by the emission factor (EF) (Vallero, 2007). Emission factors *depend* strongly on the burning conditions (Sinha et al., 2003; Ward et al., 1996; Yokelson et al., 2003), in particular on the combustion efficiency (CE). CE is defined as the fraction of reduced carbon in the fuel that is directly converted into $CO_2$ (Yokelson et al., 1996). Usually, emission factors are measured in laboratories under controlled burning conditions.
However, in ambient environment, combustion conditions are highly variable (Andreae & Merlet, 2001; Korontzi et al., 2003) introducing large uncertainties in global emission inventories through the impact of CE on EF. A case study (Frey & Zheng, 2002) for NOx emission estimates from the coal fired power plants with dry-bottom wall-fired boilers using low NOx burner showed that the EF for NOx can vary by factor of 4 or more within a same technology. The application of mean EF introduces uncertainties in the range of -29 % to +35 % *with* respect to mean emission estimates (Frey & Zheng, 2002;
Tang et al., 20*19).* Fuel type, fuel composition, combustion practices and technology are the main factor influencing combustion efficiency in the ambient environment (Silva & Arellano, 2017). To improve the accuracy of global inventories, a better quantification of combustion efficiency and EFs is needed.

In recent years, the availability of atmospheric composition measurements from Earth orbiting satellites has strongly improved. Sensors such as Scanning Imaging Absorption spectroMeter for Atmospheric Chartography (SCIAMACHY)
*(Bovensmann et al., 1999)* and Tropospheric Monitoring Instrument (TROPOMI) *(Veefkind et al., 2012)* deliver global datasets of multiple species. The satellite observations from SCIAMACHY have been used in combination with inverse modelling techniques to test and improve emission inventories (Konovalov et al., 2014; Mijling and van der A, 2012; Reuter

et al., 2014; Silva et al., 2013). By combining observations of different species (e.g. CO, $CO_2$, $NO_2$) information about common sources is obtained, and potentially also about emission ratios (Hakkarainen et al., 2015; Miyazaki et al., 2017;

Reuter et al., 2019; S. Silva & Arellano, 2017)**.**

In this study, measurements from the TROPOMI are used to investigate the combustion efficiency in *megacities.* TROPOMI is a push broom grating spectrometer on board of Sentinel 5 *Precursor* launched by ESA on 13 October, 2017 (Veefkind et al., 2012). We use the ratio of the TROPOMI retrieved tropospheric column of $NO_2$ and the total column of CO, which is formally not equivalent to combustion efficiency but can nevertheless serve as a useful proxy (Silva & Arellano, 2017; W.

Tang & Arellano, 2017). The reason for this is that $NO_x$ emission increases with combustion temperature, which is high during efficient combustion. In contrast, CO is a product of incomplete combustion, and is produced when combustion efficiency is low (Flagan & Seinfeld, 1988). The combination of these effects makes the $NO_2$/CO ratio highly sensitive to combustion efficiency. To correct for differences in the $NO_2$ and CO background concentrations, the enhancement ratio $\Delta NO_2/\Delta CO$ is used. Here $\Delta NO_2$ and $\Delta CO$ represent concentration increases compared with their respective backgrounds.

*The $\Delta NO_2/\Delta CO$ ratio is insensitive to atmospheric transport, as $NO_2$ and CO emissions are dispersed in a similar manner by the wind.* Therefore, the impact of transport cancels out in the ratio. Because of this, TROPOMI observed ratios close to emissions *sources* can be directly related to emission ratios. The aim of this study is to investigate the local relation between TROPOMI retrieved $\Delta NO_2/\Delta CO$ ratios and emission ratios in a quantitative manner, focusing on *megacities* showing significant concentration enhancements in the TROPOMI data. In the past studies, $NO_2$ from the Ozone Monitoring

Instrument (OMI) and CO from Measurement of Pollution in the Troposphere (MOPITT),  have been used to derive CO/$NO_2$ ratios(Silva & Arellano, 2017; W. Tang & Arellano, 2017). *MOPITT also has a SWIR channel (or near IR) and the multispectral (TIR/NIR) product, with near-surface sensitivity over some land regions, was used in both Silva & Arellano, 2017; Tang & Arellano, 2017.* TROPOMI provides a unique opportunity to measure CO and $NO_2$ using the same instrument at unprecedented high spatial resolution (7x7 $km^2$ at nadir) and daily global coverage (Borsdorff et al., 2018b; van Geffen et

al., 2019) making this instrument ideally suited for investigation of $NO_2$/CO ratios from space. Additionally, TROPOMI CO retrievals make use of the short-wave infrared, improving the sensitivity to surface emissions of CO compared to the thermal infrared sounders MOPITT and Infrared Atmospheric Sounding Interferometer (IASI). However, TROPOMI $NO_2$ retrievals are less sensitive to the lower troposphere, causing $\Delta NO_2/\Delta CO$ to be influenced by vertical sensitivity(Eskes & Boersma, 2003). We derived a correction factor to take this influence into account, as will be explained in detail in Section 2.5.

This paper is organized as follows: Section 2 provides detailed information about the TROPOMI CO and $NO_2$ retrieval, the approach used to quantify the $\Delta NO_2/\Delta CO$ column enhancement ratio over megacities, and how to relate it to the corresponding emission ratio. Results comparing satellite and emission inventories derived ratios are presented in section 3. Finally, section 4 summarizes our findings and presents the main conclusions.

## 2 Data and Method

### 2.1 TROPOMI CO retrievals

For this study, we are using the TROPOMI CO scientific beta data product provided by SRON (ftp://ftp.sron.nl/open-access-data-2/TROPOMI/tropomi/co/7_7/). The output is identical to the one of European Space Agency (ESA) 's operational data product but provides in addition the TM5 a priori profiles (http://tm5.sourceforge.net/) that are used in the retrieval. The SRON CO product also supplies more data for the early months of the mission which are not included in the operational product. Total column densities of CO [molecules/cm$^2$] are retrieved from spectral radiance measurements from the TROPOMI short wave infrared (SWIR) module at 2.3 µm using the SICOR algorithm (Landgraf, Brugh, et al., 2016). In this profile scaling algorithm, the TROPOMI observed spectra are fitted by scaling a reference vertical profile of CO using the Tikhonov regularization technique (Borsdorff et al., 2014). The reference a priori CO profile is derived from the TM5 transport model (Krol et al., 2005) as described in Landgraf (2016). The averaging kernel (A) is an essential component of the CO retrieval, which quantifies the sensitivity of the retrieved CO column to a change in the true vertical profile ($\rho_{true}$) *following Borsdorff et al., (2018c)* as

$$C_{retrieval} = A * \rho_{true} + \epsilon_{CO} \tag{1}$$

Where, $\epsilon_{CO}$ is the error in the retrieved CO columns.

### 2.2 TROPOMI NO$_2$ retrievals

The UV-Vis module of TROPOMI is used to retrieve NO$_2$ in the 405-465nm spectral range. NO$_2$ slant column densities are processed using the TROPOMI NO$_2$ DOAS software developed at KNMI (van Geffen et al., 2019). The retrieval algorithm is based on the NO$_2$ DOMINO algorithm ( Boersma et al., 2011) which has been improved further in the QA4ECV4 project (Boersma et al., 2018). The algorithm subtracts the stratospheric contribution to the slant column densities, and then converts the residual tropospheric slant column density into the tropospheric vertical density via the air mass factor (AMF). The AMF is computed using co-sampled, daily NO$_2$ a priori vertical profiles from output of the TM5-MP chemistry transport model at 1˚ x 1˚ resolution (Williams et al., 2017). AMF depends on the surface albedo, terrain height, cloud height and cloud fraction (Eskes et al., 2018; Lorente et al., 2017). We have used the offline level 2 NO$_2$ data [molem$^{-2}$] available at (https://s5phub.copernicus.eu; http://www.tropomi.eu). The TROPOMI NO$_2$ product has been successfully used in various studies so far (Griffin et al., 2019; Reuter et al., 2019). There are indications that *NO2 is biased low* by approximately 30% in the tropospheric columns because of issues with the cloud pressure and a priori NO2 profile used in the AMF calculation (Lambert et al., 2019).

### 2.3 Data Selection

We used TROPOMI CO and NO$_2$ retrievals from June to August, 2018 because of the large number of clear sky days during this period over *megacities* of our interest. Megacities are strong sources of air pollution and can readily be observed in

TROPOMI data (Borsdorff et al., 2018c). Since CO and $NO_2$ are retrieved from different instrument channels using different algorithms, the filtering criteria and spatial resolutions are also different. To facilitate data filtering, both algorithms provide a quality assurance value (qa value). The qa value for both products ranges from 0 (no data) to 1 (high quality data)

For our data analysis, we selected $NO_2$ retrievals with qa values equal or larger than 0.75, indicating clear sky conditions (Eskes & Eichmann, 2019), and CO retrievals with qa values equal or larger than 0.7, representing measurements under clear

sky conditions or the presence of low-level clouds (Apituley et al., 2018). *The application of SICOR algorithm on SCIAMACHY CO retrievals with low-level clouds increases the number of measurement and hardly disturb on the detection of CO sources* (Borsdorff et al., 2018a). CO retrievals are filtered for stripes as described in Borsdorff et al., (2018c). The CO retrieval has a factor 2 coarser spatial resolution than the $NO_2$ retrieval (7x7km2 versus 3.5x7km2). To collocate $NO_2$ and CO retrievals, we combine those $NO_2$ pixels which centres fall within a CO pixel, selecting only those pixels for which

both the $NO_2$ and CO retrievals pass the filtering criteria. The total CO column and tropospheric $NO_2$ columns are converted into the dry column mixing ratio XCO (ppb) and $XNO_2$ (ppb) using the dry air column density calculated using the collocated surface pressure data included in the CO data files as described in Borsdorff et al., (2018c).

**Table 1. Selected megacities and specifications used for emission ratio quantification**

| City | Centre (Latitude, Longitude) | Radius of core city (km) | Radius outskirt (km) | Radius background (km) | Upwind area Δlat , Δlon (°) |
|---|---|---|---|---|---|
| Tehran | 35.68, 51.42 | 10 | 180 | 250 | 1.0, 1.0 |
| Mexico City | 19.32, -99.20 | 10 | 170 | 180 | 1.0, 1.0 |
| Cairo | 30.04, 31.23 | 10 | 135 | 180 | 1.0, 1.0 |
| Riyadh | 24.63, 46.71 | 10 | 100 | 150 | 1.0, 1.0 |
| Lahore | 31.53,  74.35 | 10 | 165 | 200 | 1.0, 1.0 |
| Los Angeles | 34.05, -118.24 | 10 | 200 | 250 | 1.0, 1.0 |

**2.4 Calculation of $NO_2$/CO**

This study focuses on the following megacities (population > 5 million): Mexico City, Tehran, Riyadh, Cairo, Lahore and Los Angeles. These six megacities are well isolated from surrounding sources and frequently experience cloud-free conditions, allowing the retrieval of a large number of XCO and $XNO_2$ data from TROPOMI. Los Angeles and Mexico City have automated air quality monitoring networks, measuring CO and $NO_2$ at different locations in the city. These

measurements are used in section 3.3 to validate the results obtained using TROPOMI. In addition, these megacities are expected to span a sizeable range in burning efficiency by including urban centres in developed (US/ Los Angeles) and developing countries( Mexico/ Mexico City, Egypt/ Cairo, Saudi Arabia/Riyadh, Pakistan/ Lahore).

The concentration gradient between the background and the city centre is used to determine the $\Delta XNO_2/\Delta XCO$ enhancement ratio. To determine this ratio, we divide each city into a core city area and a background area. *Every city has a different size and different neighbouring CO and NO2 emission sources and therefore the appropriate choice of radii for the background and outskirt areas varies between cities (detail explanation in Supplements Section 1). This is important mostly to have a significant signal from city emissions in CO and NO2. However, since the same regional definition is used for NO2 and CO, the enhancement ratio is not so sensitive to the details of the region selection.* To maximize the size of the city enhancement, we exclude the diffuse outskirt area in between the city centre and the background. For the location of the city centre we use the weighted average emission centre of $NO_2$, derived from the EDGAR emission database (Dekker et al., 2017). The derived centre coordinates, and the radii of the city core and background area are listed in Table 1. We test the robustness of the satellite-derived emission ratio using two different methods, which are explained in detail below.

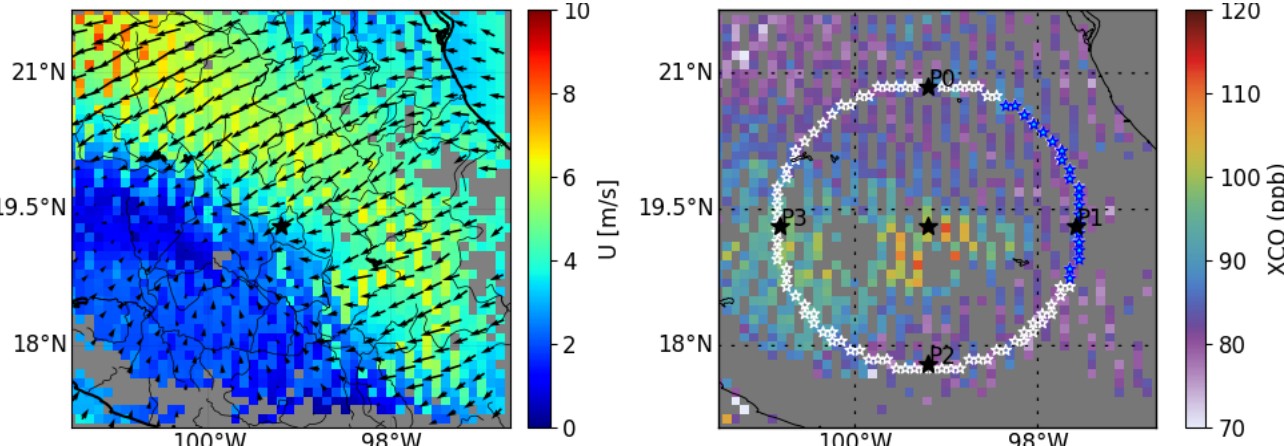

**Figure 1.** ERA interim average wind speed and direction from surface to 200m at the time TROPOMI overpasses (left ) and TROPOMI derived CO total column over Mexico City (right) for 5$^{th}$ of June, 2018. The black star represents the centre of the city. In the right panel, the white circle is the background area for Mexico City and the blue section represents the upwind background area that we selected depending upon the wind direction in the core city area. P0,P1,P2 and P3 are the points where north, east, west and south wind directions intersects at the inner rim of the background area.

### 2.4.1 Upwind background

To determine the upwind background (UB) column mixing ratio, we select a section of the background region that is upwind from the city centre using the average wind direction over the core city area (*Fig. 1and for detail see Fig S7*). Generally, more than 75% of all pollutants are emitted between the surface and 200m altitude (Bieser et al., 2011). Therefore, average wind speed and direction from surface to 200m altitude are derived from the ERA-interim reanalysis, provided at 0.75˚x0.75˚ and 3 hourly resolutions. The wind vector components of ERA-interim are spatially and temporally interpolated to the central coordinate of TROPOMI pixels. Using this information, daily enhancement ratios are calculated as follows.

$$\Delta XNO_2 = XNO_{2_{city}} - XNO_{2_{background}} \qquad (2)$$

$$\Delta XCO = XCO_{city} - XCO_{background} \qquad (3)$$

$$Ratio = \frac{\Delta XNO_2}{\Delta XCO} \tag{4}$$

The background area might contain free tropospheric $NO_2$ from lightning and convectively lofted surface $NO_2$ from elsewhere. However, these contributions vary on scales that are usually large compared with the scale of a city. Therefore,
the calculated $\Delta XNO_2$ and $\Delta XCO$ enhancements are caused predominantly by emissions from the city.

### 2.4.2 Plume Rotation

The daily TROPOMI-observed city images are rotated in the direction of wind using the city centre as the rotation point to align each CO and $NO_2$ plume in upwind-downwind direction (Pommier et al., 2013). Rotated images for June to August 2018 are averaged together *(see FigS8)*. $\Delta XNO_2$ and $\Delta XCO$ are determined by subtracting the average of the first quartile
$XNO_2$, XCO values in a 100 km x 20 km region upwind from the city centre from the average of the fourth quartile $XNO_2$, XCO values in a 100 km x 20 km region downwind from the city centre. Finally, the enhancement of $XNO_2$ and $XCO$ is calculated as described in Eq. (5) and the enhancement ratio is derived by using Eq. (4).

$$downwind - upwind \; difference = Vd - Vu =$$

$$\frac{\sum_{i=1}^{n \; downwind}(X \geq 75^{th} \; percentile)}{n \; downwind} - \frac{\sum_{i=1}^{n \; upwind}(X \leq 25^{th} \; percentile)}{n_{upwind}} \tag{5}$$

where, $n_{downwind}$ = number of observation $\geq 75^{th}$ percentile, $n_{upwind}$ = number of observation $\leq 25^{th}$ percentile

### 2.5 NO$_2$/CO emission ratio

Local TROPOMI derived ratios in column abundance are compared with emission ratios derived from the Emission Database for Global Atmospheric Research (EDGAR v4.3.2) at 0.1˚ x 0.1˚ spatial resolution for the most recent year of 2012 and the database provided by Monitoring Atmospheric Chemistry and Climate and CityZen (MACCITY), for 2018 available at 0.5˚ x 0.5˚ resolution (Granier et al., 2011). MACCity has been re-gridded to a spatial resolution of 0.1˚ x 0.1˚ assuming a uniform distribution of the emissions within each 0.5˚ x 0.5˚ grid box. Both emission inventories contain total emissions of
NOx and CO. NOx emissions are converted into $NO_2$ by dividing NOx by the conversion factor of 1.32. This conversion factor is based on Seinfeld and Pandis (2006) and represents urban plumes at 13.30 local time. The emission ratio of $NO_2$ and CO ($E_{NO2}/E_{CO}$) is calculated from total emissions (sum of all processes) within the core city area, for the EDGAR and MACCity emission inventories.

To compare TROPOMI to inventory derived ratios, the $NO_2$ tropospheric column has to be corrected for its limited
atmospheric residence time. The CO lifetime is long enough compared with the transport time out of the city domain to be neglected. In addition, we need to account for differences in the vertical sensitivity of TROPOMI to $NO_2$ and CO, as quantified by their respective averaging kernels (A) shown in Fig. 2. To compare TROPOMI to EDGAR and MACCity, we formulate a relationship between the emission ratio ($E_{NO2}/E_{CO}$) and the column enhancement ratio ($\Delta XNO_2/\Delta XCO$) taking

into account the combined effect of atmospheric transport, chemical loss and the averaging kernel. This relationship is as follows (see Appendix A for its derivation).

$$\frac{E_{NO2}}{E_{CO}} = \frac{\Delta XNO_2}{\Delta XCO} \cdot \frac{\left(\frac{U}{lx} + K[OH]\right)}{\frac{U}{lx}} \cdot \frac{1}{(1 - A_{influence})} \tag{6}$$

Where, U is the is the 200m wind speed (ms⁻¹), lx is diameter of the city centre (m), K is the rate constant of the reaction of $NO_2$ with OH of $2.8e^{-11}\left(\frac{T}{300}\right)^{-1.3}$ cm³ molecule⁻¹ s⁻¹ (Burkholder et al., 2015). T (K) and OH (molecule cm⁻³) are respectively the boundary layer average temperature and OH concentration and $A_{influence}$ is the influence of the averaging kernel on $\Delta XNO_2/\Delta XCO$ (see section 3.2).

OH, CO and $NO_2$ fields from the Copernicus Atmospheric Monitoring Service (CAMS) real time are used to account for the impacts of chemical loss and the averaging kernel. The CAMS data, at 0.1˚ x 0.1˚ and 3 hourly resolutions are spatially and temporally interpolated to the TROPOMI footprints. CAMS CO and $NO_2$ vertical mixing ratio profiles are converted into vertical column densities using ERA Interim reanalysis surface pressure. For CO, the TROPOMI data provide column A's from the surface to the top of atmosphere. For $NO_2$, tropospheric A is derived using the air mass factor for the troposphere as fraction of the total column (Boersma et al., 2016). For further details see Appendix B.

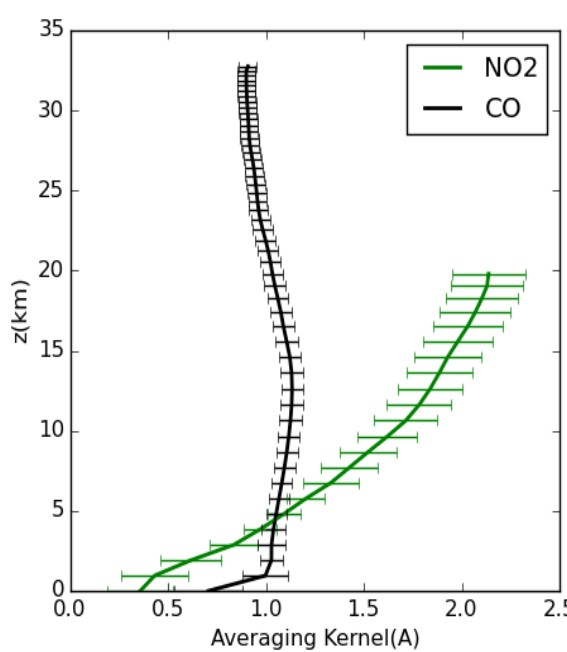

**Figure 2.** TROPOMI total CO and tropospheric $NO_2$ column averaging kernel (A) for June 1[st], 2018 over Mexico. The error bars represents the standard deviation of the mean A at each vertical level.

## 2.6 Uncertainty

To quantify the uncertainty in TROPOMI-derived $\Delta XNO_2/\Delta XCO$ ratios for the plume rotation method, we use the error propagation method of Pommier et al.,(2013) and boot strap for the upwind background, as explained further below.

### 2.6.1 Bootstrapping

The boot-strapping method is a statistical resampling method, used here to calculate the uncertainty in the daily enhancement ratio of $\frac{\Delta XNO_2}{\Delta XCO}$. The first step is to generate a new set of samples by drawing a random subset with replacement from the full dataset of N daily $\frac{\Delta XNO_2}{\Delta XCO}$ ratios. The subset has the same number of samples as the full dataset, from which a mean ratio is

calculated. This procedure is repeated a thousand times for each city. Finally, the standard deviation of the resulting ratios is taken and used to represent the uncertainty in daily $\frac{\Delta XNO2}{\Delta XCO}$.

### 230  2.4.1   Error propagation

To calculate the uncertainty in $\frac{\Delta XNO2}{\Delta XCO}$ by error propagation, we first determine the uncertainty in the enhancements $\Delta XNO_2$ and $\Delta XCO$, which are derived from the uncertainty in the mixing ratios upwind and downwind of the source as follows

$$\sigma_{\Delta X} = \sqrt{\left(\frac{\sigma_{upwind}}{\sqrt{n_{upwind}}}\right)^2 + \left(\frac{\sigma_{downwind}}{\sqrt{n_{downwind}}}\right)^2} \qquad (7)$$

where, X is $XNO_2$ or XCO.

235  Here, we assume that the upwind and downwind uncertainties are independent. The uncertainty for the column enhancement is:

$$\sigma_{ratio} = \left(\sqrt{\left(\frac{\sigma_{\Delta NO_2}}{\Delta XNO_2}\right)^2 + \left(\frac{\sigma_{\Delta CO}}{\Delta XCO}\right)^2}\right) * \frac{\Delta XNO_2}{\Delta XCO} \qquad (8)$$

## 3. Results and Discussion

### 3.1 Detection of $NO_2$ and CO pollution over megacities

240  The collocated TROPOMI $XNO_2$ and XCO data have been averaged for June to August 2018, for domains of 500 x 500 km$^2$ centred around the selected *megacities* as described in section 2. Results are shown in Fig. 3 for Mexico City and Cairo. The enhancements of XCO and $XNO_2$ over Mexico City and Cairo are clearly separated from the surrounding background areas and are prominent in several o*verpasses* of TROPOMI (Fig. S9). This demonstrates that a relatively short data averaging period is sufficient for TROPOMI to detect hotspots of CO pollution at the scale of large cities, compared to instruments 245  such as IASI and MOPITT. The orography surrounding Mexico City causes trapping of pollutants facilitating detection by TROPOMI. The longer *lifetime* of CO compared to $NO_2$ causes the urban influence of CO to be propagated further in westward direction. As can be seen in Fig. 3 the retrieved XCO and $XNO_2$ signals of emissions from Mexico City and Cairo correlate quite well with each other, confirming that it should be possible to obtain useful information about burning efficiency by studying$\frac{\Delta XNO2}{\Delta XCO}$. An industrial area is located to the east of Cairo (29.797351N, 32.148266 E), showing a clear 250  enhancement in $XNO_2$ but not in XCO (Fig. 3c and d). It demonstrates that variations in the column enhancement ratio can already be seen by eye comparing TROPOMI retrieved XCO and $XNO_2$ images.

The collocated TROPOMI $XNO_2$ and XCO data have been averaged for June to August 2018, for domains of 500 x 500 km$^2$

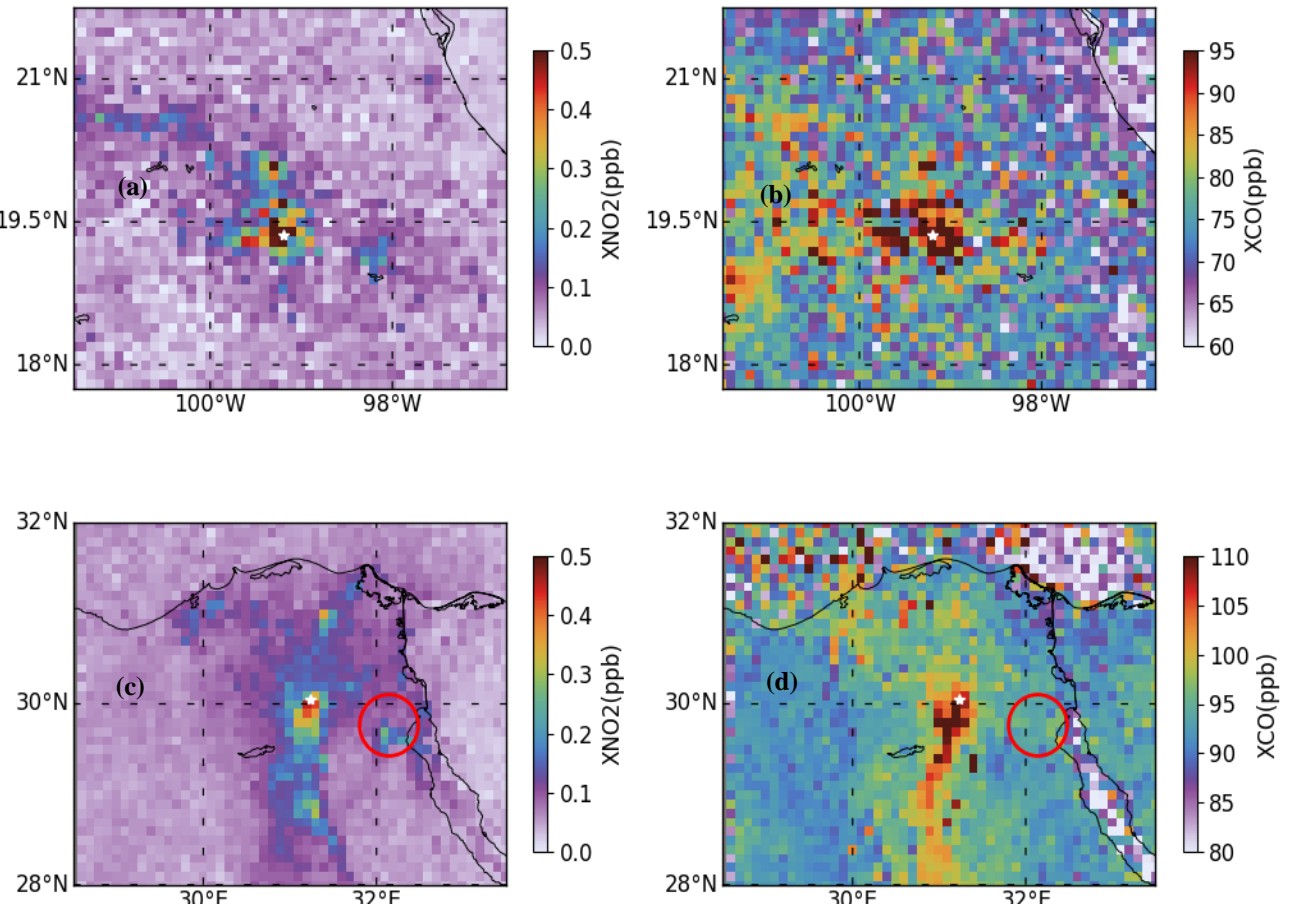

**Figure 3.** Collocated TROPOMI retrieved $XNO_2$ (left) and XCO (right) data over Mexico (top) and Cairo (bottom) averaged for June to August, 2018. De-striping is applied to CO total columns (Borsdorff et al., 2018c) and CO and $NO_2$ retrievals have been re-gridded to $0.1°x0.1°$. The white stars represent the centres of Mexico City and Cairo, respectively. The red circle in panels c) and d) points to an industrial area eastward of Cairo.

## 3.2 Comparison between TROPOMI and inventory derived ratios

In this subsection, we attempt to compare TROPOMI-derived $NO_2$/CO column enhancement ratios to emission ratios from EDGAR and MACCity for the six selected *megacities* (see Fig. 4). As explained in section 2, column enhancement ratios from TROPOMI are obtained using the upwind background (UB) and plume rotation (PR) methods. These estimates differ by 5 to 20 % across the six cities, providing an initial estimate of the accuracy at which the column enhancement ratio can be derived (see Table S1 for details). The EDGAR and MACCity inventories show a substantial variation in emission ratios between cities, with relatively high emission ratios for Riyadh and the lowest for Lahore. TROPOMI-derived $\Delta XNO_2/\Delta XCO$

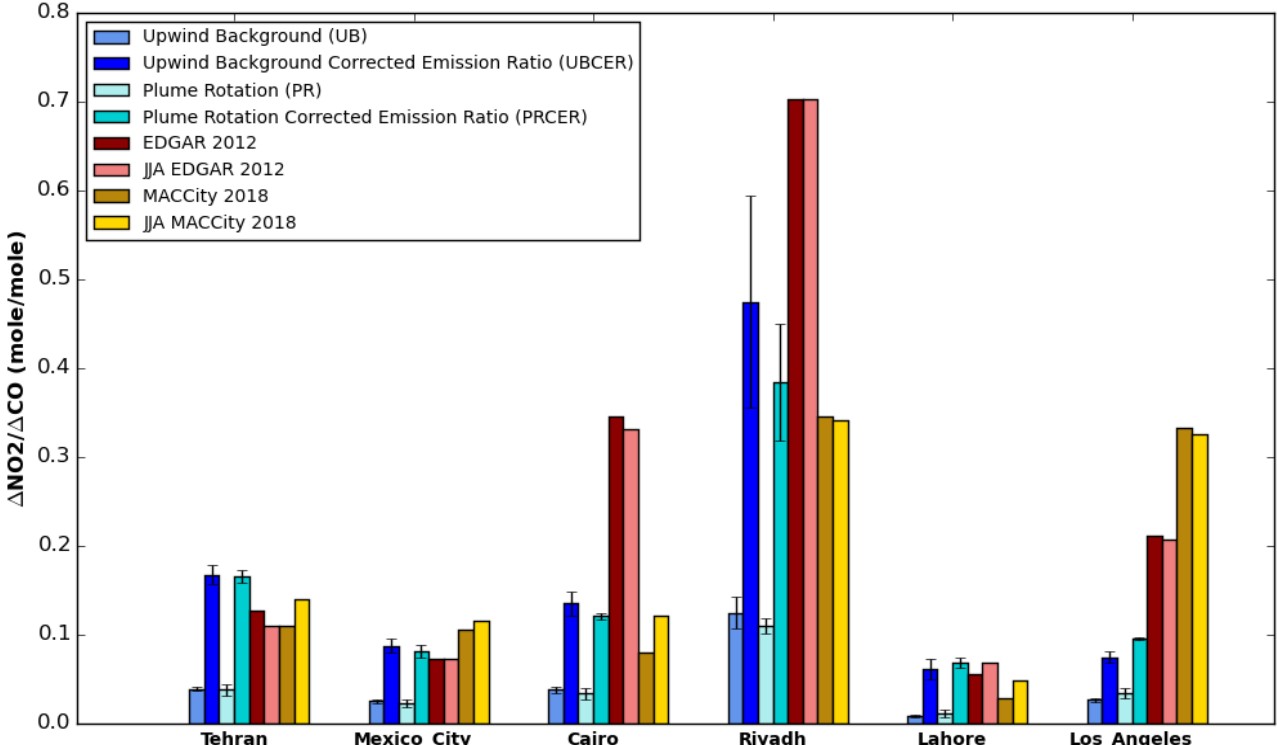

**Figure 4.** Comparison of TROPOMI-derived $\Delta NO_2/\Delta CO$ enhancement ratios, calculated using different methods shown in blue shades, to corresponding emission ratios from the EDGAR (red shades) and MACCity (yellow shades) emission inventories for six *megacities*. The dark solid shades for emission inventories represent the annual average inventory derived ratio whereas faded shades represents the June to August average inventory derived ratio. Error bars represent 1σ uncertainties calculated using boot strapping (upwind background) and error propagation (plume rotation method). The upwind background corrected emission ratio (UBCER) and Plume rotation corrected emission ratio (PRCER) account for the impact of photochemical $NO_2$ removal and the averaging kernel.

column enhancement ratios for the UB and PR methods show similar patterns as EDGAR and MACCity with Pearson correlation coefficients of 0.85 and 0.7 respectively (Fig. S10). However, inventory-derived emission ratios are clearly larger than TROPOMI-derived column enhancements ratios by 60 to 85%, explained largely by the impact of the limited NO2 lifetime and the averaging kernel, as will be discussed further after explaining the differences between EDGAR and MACCity.

Emission ratios from MACCity are lower than from EDGAR by 10 to 75%, except for Los Angeles and Mexico City. To understand the differences in emission ratios between MACCity and EDGAR, we selected two cities, Cairo and Mexico City, which present the largest and smallest differences in emission ratio. The CO and $NO_2$ emissions are categorized into seven sectors: agriculture, residence, energy, industries, transportation, shipping and waste treatment. Sectors are compared that contribute most to the total emission. In the case of Cairo and Mexico City these are the transportation, industries, energy and resident sectors (Fig. S11 a and b). For Cairo, the total CO emission is lower in EDGAR than in MACCity by a

factor 2, whereas the total $NO_2$ emission is 10% higher in EDGAR. This results in an emission ratio that is higher by a factor 3. The largest discrepancy between EDGAR and MACCity CO emission is due to the resident sector followed by energy. For $NO_2$, the energy, transportation and resident sectors explain most of the difference between EDGAR and MACCity. In

Mexico City, EDGAR total CO and $NO_2$ emission are both higher by a factor 2 compared to MACCity, cancelling out in the ratio leading to the best agreement of all selected *megacities*. However, it is complicated to identify the main factors explaining the differences between EDGAR and MACCity at the sector level due to the combined influence of differences in activity data, emission factors and the methods used to disaggregate country totals. To understand the disaggregation of emission in EDGAR and MACCity, we compared the country total CO and $NO_2$ of Mexico/ Mexico City and Egypt/ Cairo.

The comparison shows that EDGAR and MACCity country CO total and $NO_2$ total of Mexico shows a small differences (~12%) whereas in Mexico city the difference is about factor of 2 (Fig. S11c). For Egypt, EDGAR and MACCity CO total shows the similar differences as Cairo whereas EDGAR $NO_2$ country total emission is lower by factor 2 (Fig. S11d). This shows that EDGAR attribute CO and NO2 emission to the city and MACCity smears them out over the country.

The difference between satellite-derived column enhancement ratios and inventory-based emission ratios can be explained in

part by the relative short lifetime of $NO_2$, reducing columnar $NO_2$/CO ratios compared to the emissions. In addition, the sensitivity to the planetary boundary layer is smaller for $NO_2$ than for CO TROPOMI measurements, reducing the

satellite observed column enhancement ratio further. Taking these influences into account using Eq. (6) leads to the Upwind Background Corrected emission ratio (UPCER) and Plume rotation Corrected Emission Ratio (PRCER) in Fig. 4,

which have been calculated on a daily basis before averaging over the full period. Due to the short lifetime of OH, its concentration depends strongly on the local photochemical conditions (de Gouw et

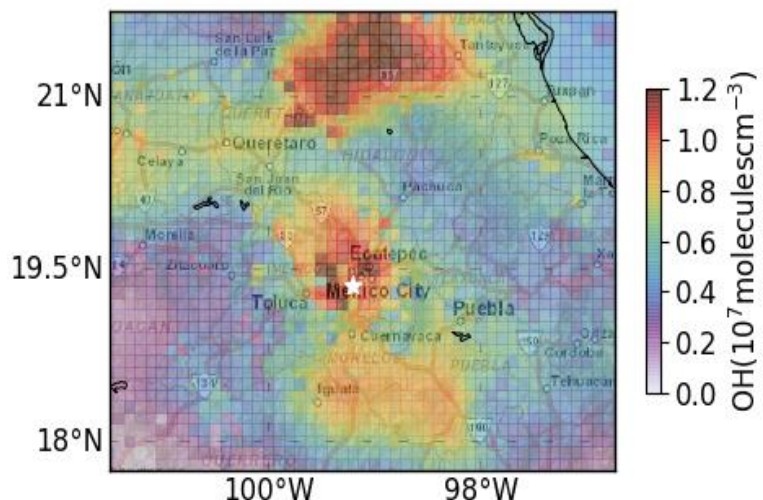

**Figure 5.** The boundary layer average OH concentration at the time of TROPOMI overpasses during June –August, 2018 over Mexico City. The white star represents the centre of Mexico City.

al., 2019). Therefore, to account for the local lifetime of $NO_2$, we need an estimate of the OH that is representative for the

photochemical conditions inside cities. Figure 5 shows the boundary layer OH concentration at the time TROPOMI overpasses from CAMS for Mexico City, averaged over June-August, 2018. The Fig. 5 shows a clear enhancement of OH in the city centre, confirming that the spatial resolution of CAMS is sufficient to resolve urban influences on OH in megacities. UB and PR column enhancement ratios increase by 60 to 85 %, when accounting for the $NO_2$ lifetime (see Table S1). The boundary layer OH concentrations and mean wind speeds for the six cities are listed in Table 2.

The impact of differences between the $XNO_2$ and XCO averaging kernels is calculated using vertical profiles of $NO_2$ and CO taken from CAMS. These profiles were used to calculate $XNO_2$ and XCO using either the TROPOMI A's or A's replaced by identity matrices. The relative difference $A_{influence} = \frac{(Without\ A - with\ A)}{Without\ A} \cdot 100\%$ quantifies the impact of differences between the averaging kernels (see Appendix C for the derivation). *The CAMS daily simulated city enhancements over June to August, 2018 did not compare well with TROPOMI for CO over Tehran, Cairo, Riyadh and Lahore, possibly due to the*

*coarse resolution of CAMS (see Fig S14, S15, S16and S17). Therefore, $A_{influence}$ for Mexico and Los Angeles is determined to calculate the averaging kernel impact (Fig S12 and S13). CAMS derived enhancement ratio for Mexico City differs by 5 % compared to UB and PR but for Los Angeles the ratio (0.094) is higher by 75% compared to UB and PR (0.034). To test the accuracy of $A_{influence}$, a few days for Tehran, Cairo, Riyadh and Lahore were selected when CAMs CO and $NO_2$ enhancements did compare relatively well with TROPOMI.* For the six megacities, TROPOMI derived $\Delta NO_2/\Delta CO$ ratios are

10 to 15 % lower than the 'ideal' $\Delta NO_2/\Delta CO$ ratio that would be measured if both retrievals had uniform vertical sensitivities, i.e. every molecule in the column receives equal weight. Details about the selected days and calculated corrections for each city are listed in Table S2.

After correction, UBCER and PRCER for Tehran and Mexico City are close to EDGAR and MACCity (10 to 25%). This confirms that the emission factors for Mexico City is well represented in the EDGAR and MACCity emission inventories.

The difference between corrected and uncorrected ratios in Fig. 4 highlights the importance of the correction, in particular the influence of OH, for assessing emission ratios using TROPOMI. For Cairo the correction also reduces the difference between TROPOMI and the emission inventories, although the EDGAR ratios remain higher by about 65% for Cairo than UBCER and PRCER. For MACCity, the emission ratios are close to TROPOMI derived UBCER and PRCER for Cairo (within 20%), pointing to a more accurate representation of emission ratios in MACCity than in EDGAR. However, for

Riyadh UBCER and PRCER are close to MACCity (~10 -20 %) whereas EDGAR is higher by 35%. Similarly, for Lahore, PRCER is close to EDGAR ratio whereas, MACCity is lower by factor of 2.5. For Los Angeles, the ratios from MACCity and EDGAR are higher by 70 % and 50 % respectively than UBCER and PRCER after correction, suggesting poorer burning conditions than represented by the emission inventories. To further investigate this discrepancy for Los Angeles, we included the Hemispheric Transport of Air pollution version 2 (HTAP-v2) emission inventories for 2010 in the comparison. HTAP-v2

has a resolution of 0.1˚ x 0.1˚ and makes use of emission estimates from the Environmental Protection Agency (EPA) for the USA (Janssens-Maenhout et al., 2015). The HTAP-v2 derived emission ratio over Los Angeles is 0.074, which is close to UBCER and PRCER (within 20 %). This result provides further confidence in TROPOMI derived emission ratio. However, different sources of uncertainty play a role as discussed further below.

To account the effect of seasonality on the annual average EDGAR ratio, *we quantify the seasonal correction factor using*

*EDGAR v4.3.2 2010 since monthly data for EDGAR 2012 is not available. June to August (JJA) EDGAR ratio reduces by 12.5 % for Tehran whereas <5% for Cairo, Riyadh, Mexico City and Los Angeles. For Lahore, ratio increased by 24.3 % compared to annual average EDGAR ratio (Figure 4 and Figure S18). The JJA MACCity 2018 ratio is higher by 27.0 % for*

**Table 2. Average wind speed and boundary layer CAMs OH concentration for June- August, 2018, used to correct for the limited lifetime of NO$_2$.**

| Cities | Mean wind speed (kmh$^{-1}$) | Mean OH concentration (10$^7$ moleculescm$^{-3}$) | Conversion factor |
| --- | --- | --- | --- |
| Tehran | 12.7 | 1.77±0.15 | 1.23±0.005 |
| Mexico City | 11.3 | 1.0±0.1 | 1.27±0.009 |
| Cairo | 16.5 | 1.85±0.14 | 1.24±0.0029 |
| Riyadh | 21.1 | 1.6±0.2 | 1.35±0.007 |
| Lahore | 7.0 | 1.3±0.2 | 1.19±0.006 |
| Los Angeles | 15.1 | 1.2±0.1 | 1.25±0.006 |


*Tehran, 10 % for Mexico City, 50 % Cairo and 71 % for Lahore. The JJA MACCity ratio is close to UBCER and PECER (within 10 %) for all the cities except Los Angeles. EDGAR and MACCity do not agree with the effect of seasonality on the emission and comparison of seasonal ratio might result uncertainty in inventory derived ratio.*

The ozone concentration and the photolysis rate impact the partitioning of NO and NO$_2$ (Jacob, 1999) influencing the
applied conversion factor of 1.32. To further investigate the uncertainty introduced by this factor, we analysed CAMS surface NO and NO$_2$ at the time of the TROPOMI overpasses (see Table 2). The CAMS-derived conversion factor varies <10 % compared with the standard value of 1.32, introducing a <10 % uncertainty in the inventory derived emission ratio. However, given the uncertainty in the CAMS simulated urban NO, NO$_2$ and OH concentrations (Huijnen et al., 2019) the actual uncertainty is probably higher. Additionally, TROPOMI underestimates NO$_2$ column by 7 % to  29.7 % relative to
MAX-DOAS ground based measurement in European cities (Lambert, et al., 2019). *However, since we don't know yet how representative this estimate is for the cities that we study so, the impact of the bias is accounted as an additional the source of uncertainty of 25% of the TROPOMI inferred NO2/CO ratio (see Table S3). We calculated the wind direction and wind speed at different height i.e. 200m to 1000m and the ratio changes<10 % for all the cities (FigS19 and S20). The initial uncertainty for CAMS OH was ±50 % (Huijen et al., 2019). The bootstrapping method show that the concentration of OH*
*varies from 8.0 – 15 % for six different megacities resulting similar uncertainty to the TROPOMI derived emission ratio. If the CAMS overestimate OH concentration systematically, the TROPOMI derived emission ratio will decrease. To estimate the effect of predefined areas as background, we simultaneously increase the outskirt and background radius by 10 km for all the cities for four times. The effect is about 20 % for Riyadh whereas for other cities, the effect is < 12 % (Fig S21). The bias in S5P TROPOMI NO2 retrievals has the largest contribution for the total uncertainty on satellite derived emission*
*ratio. The wind direction and speed, boundary layer OH concentration, A$_{influence}$ correction and the predefined background setting contributes the negligible uncertainty on the TROPOMI derived emission ratio. The total uncertainty calculated*

*using error propagation method for TROPOMI derived emission ratio ranges from 27 to 35 % and the detail is provided in Table S3 (see supplements).*

We also acknowledge that our treatment of the photochemical removal of $NO_2$ is simplified. In reality, $NO_2$ is influenced by several other factors including meteorological parameters such as temperature, wind speed and radiation (Lang et al., 2015; Romer et al., 2018), causing the formation and loss of $NO_2$ to vary spatially and temporally. In the corrected ratio, we only consider the first order loss of $NO_2$ by OH forming $HNO_3$. Several studies show that in cities surrounded by forested areas, loss of $NO_2$ through the formation of alkyl and multifunctional nitrates ($RONO_2$) can play a more important role than nitric acid production (Browne et al., 2013; Farmer et al., 2011; Romer Present et al., 2019; Sobanski et al., 2017). In addition, secondary production of CO from VOC oxidation may play a role. However, this only affects our ratios if it changes the CO gradient between the city and the background. Hence, to further improve the accuracy of TROPOMI supported evaluation of emission ratios a more sophisticated treatment of urban photochemistry is required.

### 3.3 Validation using ground based measurements

To further evaluate TROPOMI's ability to quantify burning efficiencies, TROPOMI derived $\Delta XNO_2/\Delta XCO$ ratios have been compared with ground-based measurements from Mexico City and Los Angeles. For this purpose, twenty ground-based stations in Mexico City with hourly measurements of CO and $NO_2$ have been selected from the AIRE CDMX network (http://www.aire.cdmx.gob.mx/). Similarly, for Los Angeles twelve ground based stations from South Coast Air Quality Management District (AQMD) monitoring network (www.aqmd.gov/) have been selected. For the details of the names and locations of these sites see Table S4. For Mexico City, data were only available for June 2018. For Los Angeles, data for the June to August 2018 period were used but the periods 25 July to 11 August and 17 to 26 August were excluded to avoid the influence of wild fires on the observed urban pollution level.

The validation results are presented in Fig. 5 for spatially averaged, hourly CO and $NO_2$ measurements for Mexico City and Los Angeles collected during noon (12:00 to 14:00 local time). To determine the enhancement in CO and $NO_2$ due to local emissions for each ground-based station, the $5^{th}$ percentile of hourly CO and $NO_2$ measurements

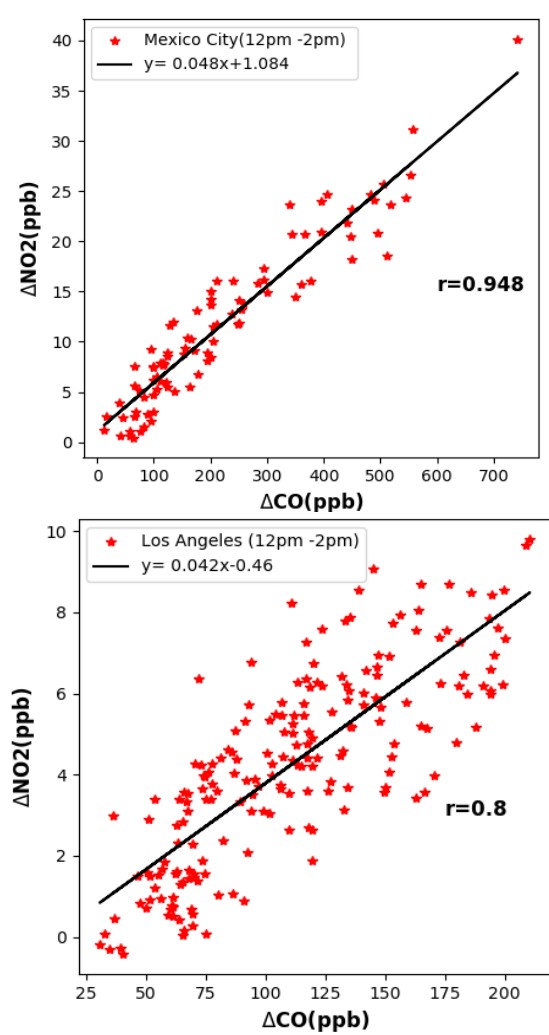

**Figure 6.** Ground based $\Delta NO_2$ versus $\Delta CO$ for Mexico (top) and Los Angeles (bottom). The red dots represent spatially averaged hourly measurements collected during the day (12:00 to 14:00 local time)

is used as background. $\Delta CO$ and $\Delta NO_2$ enhancements for individual monitoring stations are calculated as $\Delta X = X_{individual} - X_{background}$.

To compare with TROPOMI, all measurement sites are spatially averaged.

Ground based $\Delta CO$ and $\Delta NO_2$ at Mexico City and Los Angeles are strongly correlated with a Pearson correlation coefficient of r = 0.95 and 0.80 respectively, confirming that the observed signals reflect $NO_2$ and CO emissions from common sources.

The slope of the regression line for Mexico City amounts to 0.048, which is 45 % higher than the TROPOMI derived column enhancement ratio using the UB and PR method. The $\Delta NO_2/\Delta CO$ ratio that is observed at ground level is likely influenced less by photochemical removal of $NO_2$ than the TROPOMI retrieved columns, and therefore closer to the inventory derived ratio, consistent with our results. This comparison suggests that removal of $NO_2$ reduces the ratio in ground-based measurements by 35 % compared to EDGAR and MACCity. Overall, the emission ratios in EDGAR and MACCity for

Mexico City are consistent with both the ground-based measurements and TROPOMI, i.e. within the uncertainty of introduced by the chemical removal of $NO_2$.

For Los Angeles, the regression slope is 0.042, which is 20% larger than the TROPOMI derived column enhancement ratios using the UB and PR method. However, the EDGAR and MACCity ratios are higher by a factor 5 compared to the $\Delta NO_2/\Delta CO$ ratio observed at ground level. The ground-based measurements point to similar ratios for Mexico City and Los

Angeles, confirming the HTAP-v2 supported TROPOMI finding that the emission ratio in EDGAR and MACCity is too high for Los Angeles. Therefore, the ground-based measurements for Los Angeles provide independent support for the TROPOMI derived ratios pointing to poorer burning conditions in Los Angeles than indicated by the emission inventories, and confirm the value of TROPOMI for monitoring the burning efficiency of megacities.

## 4. Conclusion

In this study, we investigate the use of TROPOMI XCO and $XNO_2$ retrievals for monitoring the burning efficiency of fossil fuel use in megacities. To improve the accuracy of the global emission inventories, the burning efficiency and emission factor is quantified using collocated XCO and $XNO_2$ enhancements over the megacities Tehran, Mexico City, Cairo, Riyadh, Lahore, and Los Angeles. TROPOMI is well capable of detecting XCO and $XNO_2$ enhancements over these megacities with relatively short averaging time and shows the expected spatial correlation.

TROPOMI derived column enhancement ratios have been compared with emission ratios from EDGAR and MACCity. The TROPOMI derived column enhancement ratios are strongly correlated with EDGAR and MACCity inventory derived emission ratios (r = 0.85 and 0.7) showing the highest emission ratio for Riyadh and the lowest for Lahore. This shows that Lahore has the poorest burning efficiency whereas over Riyadh, fossil fuel burning is the most efficient of all megacities that were analysed.

The impact of the short $NO_2$ lifetime and differences in the vertical sensitivity of the TROPOMI XCO and $XNO_2$ retrieval on the $\Delta NO_2/\Delta CO$ enhancement ratio has been quantified. Correcting for these factors significantly improves the agreement

between ratios derived from TROPOMI and emission inventories. The comparison indicates that emission ratio in MACCity and EDGAR is well represented for Mexico City and Tehran. EDGAR emission ratios for Lahore are better quantified whereas MACCity emission ratio over Cairo and Riyadh are close to TROPOMI derived emission ratio. However, emission

ratios in MACCity and EDGAR remain higher by 50 to 70 % for Los Angeles. *The total uncertainty on TROPOMI derived emission ratio ranges from 27 to 35 %. The bias in S5P TROPOMI NO2 retrievals accounts for the major contribution for the uncertainties in the TROPOMI derived emission ratio.*

TROPOMI derived $\Delta XNO_2/\Delta XCO$ column enhancement ratios for Mexico City and Los Angeles have been validated using ground-based measurement from local air quality monitoring networks. For Mexico City, the enhancement ratio derived

from ground-based measurements is consistent with EDGAR, MACCity and TROPOMI derived emission ratio. *CAMS derived enhancement ratio over Mexico City differs by 5 % compared to UB and PR.* For Los Angeles, TROPOMI derived enhancement ratios are consistent with the ground-based measurements as well as the HTAP-v2 inventory based on EPA statistics, whereas EDGAR and MACCity-derived emission ratios appear to be overestimated by a factor 3. *Similarly, CAMS derived enhancement ratio for Los Angeles is higher by 75 % in contrast to UB and PR.* This demonstrates the potential of

TROPOMI data for monitoring burning efficiency and evaluating emission inventories.

*Data availability:* TROPOMI NO$_2$ and CO data are used for this paper. These data can be downloaded from https://s5phub.copernicus.eu; http://www.tropomi.eu and ftp://ftp.sron.nl/open-access-data-2/TROPOMI/tropomi/co/7_7/. Ground based network data for Mexico and Los Angeles can be downloaded from http://www.aire.cdmx.gob.mx/ and

www.aqmd.gov/ respectively. EDGAR v4.3.2, MACCity and HTAP-v2 data are available at https://eccad3.sedoo.fr/. CAMS data can be downloaded from https://apps.ecmwf.int/datasets/data/cams-nrealtime/levtype=ml/.

*Author Contributions:* S.L performed data analysis, interpretation and writing paper. SH supervised the study. SH, FKB, IA, MK, HACDG, AJD discussed the result. TB and AL provided modified Copernicus Sentinel data 2018 CO data. All the authors commented on the manuscript and improve it.

*Competing interests:* The authors declare that they have no conflict of interest.

*Acknowledgements* : We would like to thank the team that has realized the TROPOMI instrument, consisting of the partnership between Airbus Defence and Space Netherlands, KNMI, SRON, and TNO, commissioned by NSO and ESA. Sentinel-5 Precursor is part of the EU Copernicus program, and Copernicus Sentinel data 2018 has been used. This research is funded by the NWO GO program (grant 2017.036). We thank to T.B and A.L for providing the modified Copernicus

Sentinel data 2018 CO data. T.B. and A.L. are funded by the TROPOMI national programme through NSO. We thank SurfSara for making the HPC platform Cartesius available for computations through computing grant 17235. We would like to thank South Coast Air Quality Management District (AQMD) monitoring network and Calidad del aire for the free use of air quality data.

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

Appendix A

Derivation of Eq. (6)

665    For CO :

The mass balance equation for CO is

$$\frac{d\Delta XCO}{dt} = Emission - loss\ by\ transport$$

$$\frac{d\Delta XCO}{dt} = E_{CO} - \frac{U}{lx}\Delta XCO$$

In steady state $\frac{d\Delta X_{CO}}{dt}$ is zero.

$$E_{CO} = \frac{U}{lx}\Delta XCO$$

where, $\Delta XCO$ is the enhancement of CO in the city in ppb, U is the wind speed in ms$^{-1}$ , lx is the diameter of the city in meter (m).

670    For NO$_2$:

The mass balance equation for NO$_2$ is :

$$\frac{d\Delta XNO_2}{dt} = Emission - loss\ by\ the\ transport - chemical\ loss$$

$$\frac{d\Delta XNO_2}{dt} = E_{NO_2} - \frac{U}{lx}\Delta XNO_2 - \frac{\Delta XNO_2}{\tau}$$

In the steady state, $\frac{d\Delta X_{NO2}}{dt}$ is zero and $\tau$ is $\frac{1}{K[OH]}$, K is the rate constant reaction of NO$_2$ with OH, $2.8e^{-11}\left(\frac{T}{300}\right)^{-1.3}$ cm$^3$ molecules$^{-1}$second$^{-1}$ (Burkholder et al., 2015), T in kelvin and OH ( molecules cm$^{-3}$) is the average boundary layer concentration.

675    $E_{NO_2} = \Delta XNO_2\left(\frac{U}{w} + \frac{1}{\frac{1}{K[OH]}}\right)$

where, $\Delta XNO_2$ is the enhancement of NO$_2$ in the City in ppb, U is the wind speed in ms$^{-1}$, lx is the diameter of the city in meter(m).

Ratio:

$$\frac{E_{NO_2}}{E_{CO}} = \frac{\Delta XNO_2}{\Delta XCO}\cdot\left(\frac{\frac{U}{lx} + K[OH]}{\frac{U}{lx}}\right)$$

Influence of averaging kernel:

$$\frac{E_{NO2}}{E_{CO}} = \frac{\Delta XNO_2}{\Delta XCO}\frac{\left(\frac{U}{lx} + K[OH]\right)}{\frac{U}{lx}}\cdot\frac{1}{(1 - A_{influence})}$$

680    Where, A$_{influence}$ is the influence of the averaging kernel on $\Delta XNO_2/\Delta XCO$

**Appendix B**

Derivation of Tropospheric Averaging kernel (A) for $NO_2$ as described by Eskes et al., (2018)

$$A_{trop} = \left(\frac{M}{M_{trop}}\right) * A_{total} \quad \left(l \le l_{tp}^{TM5}\right)$$

$$A_{trop} = 0, \qquad\qquad \left(l > l_{tp}^{TM5}\right)$$

where, M is the total mass factor and $M_{trop}$ is the air mass factor for the troposphere, $l_{tp}^{TM5}$ is the TM5 tropopause layer index.

**Appendix C**

$$Without\ A \quad = \frac{\Delta NO_{2_{CAMS}}}{\Delta CO_{CAMS}}$$

$$NO2_{new\ CAMS} \quad = NO_{2_{CAMS}} * A_{No2\ TROPOMI}$$

$$CO_{new\ CAMS} \quad = CO_{CAMS} * A_{CO\ TROPOMI}$$

$$With\ A \quad = \frac{\Delta NO_{2_{new\ CAMS}}}{\Delta CO_{new\ CAMS}}$$

$$A_{influence} = \frac{(Without\ A - with\ A)}{Without\ A}.100\%$$

685     where, $NO_{2_{CAMS}}$ and $CO_{CAMS}$ is the CAMS column densities derived for NO2 and CO whereas $\Delta NO_{2_{CAMS}}$ and $\Delta CO_{CAMS}$ is the city enhancement of $NO_2$ and CO. $A_{No2\ TROPOMI}$ and $A_{CO\ TROPOMI}$ is the TROPOMI averaging kernel for $NO_2$ and CO.