# Peer review of "Quantifying burning efficiency in Megacities using NO2/CO ratio from the Tropospheric Monitoring Instrument (TROPOMI)"

_Atmospheric Chemistry and Physics, 2019_

## Referee Comment (RC1) · Anonymous Referee #1 · 13 Jan 2020

General Comments

This manuscript presents a novel application of satellite measurements of CO and NO2 to estimate regional-average burning efficiency for a number of large cities. The method is enabled by the capabilities of a relatively new satellite sensor and will likely interest many readers of ACP. With one major exception, the presented methods seem sound and the paper is generally well written. The following issues should be addressed before publication of this manuscript in ACP.

The one significant problem with the manuscript is the discussion of how the differing vertical sensitivities (column averaging kernels) of TROPOMI CO and NO2 retrievals

are handled. To quantify the impact of this effect on Delta(XNO2)/Delta(XCO) ratios, the authors introduce the variable A_influence in Eq. 6. It is unclear how this factor was derived or how it is calculated in practice; no derivation appears either in the main text or Appendices. Presumably, it somehow depends on the TROPOMI CO and NO2 averaging kernels, but these dependences are not presented. There is a paragraph on the effects of the differing averaging kernels at the bottom of p. 11 (lines 282-290), but this paragraph only adds to the confusion since nowhere does it actually refer to the variable A_influence.

In the same paragraph, the authors report that "The CAMS simulated city enhancements averaged over June to August, 2018 did not compare well with TROPOMI for CO, possibly due to the coarse resolution of CAMS. Therefore, to calculate the averaging kernel impact, a few days were selected when CAMs CO and NO2 enhancements did compare relatively well with TROPOMI." This gives the impression that the authors' method of analyzing the effects of the averaging kernel differences for CO and NO2 was based on a small number of 'cherry-picked' cases where the higher resolution of TROPOMI (compared to CAMS) was not an issue. Thus, it appears that the authors are probably underestimating the uncertainty of the averaging kernel-related error. I believe this entire issue requires more discussion and perhaps more analysis.

Specific Comments

1. The actual lifetimes of CO and NO2 should be discussed somewhere, perhaps in the paragraph that begins on p. 3, l. 75.

2. p. 4, l. 103. Rodgers (2000) does not specifically discuss this type of retrieval algorithm and is not really an appropriate reference. Mathematically, averaging kernels play a different role in optimal estimation-based methods (as described by Rodgers) and Tikhonov regularization.

3. The chosen QA threshold values (0.75 for NO2 and 0.7 for CO) would seem to allow low-clouds for CO retrievals but not for NO2 retrievals. Are scenes with clouds

excluded from this study because of the stricter QA threshold value for NO2? Clouds could have a significant impact on the TROPOMI CO column averaging kernels.

4. Conceptually, the Upwind Background and Plume Rotation methods seem to have much in common. The text in Sections 2.4.1 and 2.4.2 should somewhere discuss expected differences in the outcomes from these two methods. Are there any obvious pros and cons to each method?

5. For the Plume Rotation method, why use the first quartile upwind and fourth quartile downwind concentrations (instead of simple averages for upwind and downwind regions)?

Technical Corrections (partial list)

The numeral 2 should be subscripted in 'NO2' (in Abstract and elsewhere).

Throughout the paper, 'mega cities' and 'mega-cities' should be replaced by 'megacities.'

p. 2, l. 48. 'depends' should be 'depend'

p. 2, l. 55. 'in respect' should be 'with respect'

p. 2, l. 66. 'precursor' should be capitalized

p. 3, l. 76. 'source' should be 'sources'

p. 8, l. 229. 'over passes' should be 'overpasses'

p. 8, l., 232. 'life time' should be 'lifetime'

p. 9, l. 244. missing Delta symbols before XNO2 and XCO
* * *

---

## Referee Comment (RC2) · Anonymous Referee #2 · 6 Mar 2020

The paper presents NOx/CO emission ratios as derived from satellite observations of NO2 and CO. It demonstrates the high potential of TROPOMI for atmospheric research. The paper is generally well written. However, the details of the method lack some details, and the robustness of the results is hard to evaluate with the given information. Thus, before publication, major revisions are necessary.

The method description has to be extended. In particular, the two approaches have to be illustrated for real data rather than just for a schematic plot. In addition, the definitions of background and upwind areas have to be made consistent for all cities. If this is not possible, the authors have to discuss this in detail, motivate their choices, and

add a discussion of uncertainties due to the a-priori settings. Error bars only account for statistical day-to-day variations, but ignore the impact of a-priori settings and the CER procedure.

Detailed comments:

- Line 117: Is this bias in NO2 accounted for in your study? How would the results change if NO2 would be scaled up accordingly?

- Table 1 settings: I am puzzled by the different definitions for different cities, especially as Line 146 states that the settings are "not critical". So why are they different at all? Do you need to tune the area definitions in order to get the right results??? How do the results look like if a consistent setup is chosen for all cities? Why is the upwind area for Riyadh different for dlat and dlon by a factor of 30?

Page 6:

- the methods are explained in plain words, but not illustrated for real data. So Fig. S2 should be moved to the main text, and the background/upwind regions etc. should be marked in this plot. In addition, the ERA wind vector should be added. The rotated patterns and the percentiles used for the second approach should be provided in a separate figure.

- both methods compare columns "upwind" and "downwind" of the investigated cities. This approach requires that there *is* transport taking place. Wind speeds for Mexico City are quite low, as can also be seen in Fig. S2. So did you consider a minimum threshold for the wind speed? I expect that it would help to remove inconclusive days.

- Line 157: with Eqs 2&3, daily ratios are calculated. But how is the total ratio (shown in Fig. 3) derived? Is it the mean of all daily ratios? This is by definition different from the second approach, where first CO and NO2 are averaged and then the ratio of means is calculated. Thus, also for approach 1, the ratio of means should be taken.

- Line 196: What emission database is used by CAMS? EDGAR? MACCity? Or

something else? How far does this affect the following interpretation and discussion of CAMS OH? How do CAMS spatial patterns of CO and NO2 compare to TROPOMI? Please provide a Figure in the Supplement. Is TROPOMI CO and/or NO2 assimilated in CAMS?

- 2.6.1: The bootstrapping approach evaluates the statistical uncertainty for the results with the chosen approach. But on top, there are also other uncertainties, like systematic effects introduced by the definition of radii etc. In particular the uncertainties of the wind direction and wind speed have to be discussed as well.

- Figure 3: Please estimate the uncertainties of NO2 lifetime and AK correction and provide error bars for the CER results as well. I expect that these uncertainties are far higher (and thus more relevant) than the purely statistical bootstrap uncertainties.

- 3.2ff: Please check the discussion and conclusions (a) for NO2 probably being biased low and (b) according to the quality of CAMS emissions and the agreement between TROPOMI and CAMS

Minor comments:

- Lines 40-42: The references stating the high uncertainty of Chinese emissions are from a time period where development in China was vastly increasing. Meanwhile, NOx emissions have been reduced, and the awareness of air pollution has increased in China. I would thus assume that these high uncertainties do not hold any longer.

- Lines 60-61 Please provide refs to SCIAMACHY (Bovensmann) and TROPOMI (Veefkind).

- Line 70: Should be NOx emission.

- Line 75: Transport disperses NO2 and CO similarly, but the lifetime of NO2 is far shorter! See the different plume extents shown in Fig. S2.

- Line 117: Avoid misreading as "the bias is low", e.g. "NO2 is biased low by about

[Figure]

30%"...

- Table 1 lat/lon: Please provide consistent number of digits for lat/lon. .01° should be accurate enough.

- Fig. S1: I don't understand why there is a need for separating 4 different wind directions in the formalism; rotation matrix should work the same for all four cases!?

---

## Referee Comment (RC3) · Anonymous Referee #3 · 9 Mar 2020

This study presents new results from TROPOMI for NO2/CO emission factors that provide information about combustion efficiency on urban scales. This is an important result for understanding how well these emissions are represented in standard inventories with subsequent impacts for air quality and climate model predictions. I recommend publication after the comments from 2 other referees and some minor issues from me are addressed.

Following the comment of Ref.#1 in addressing the different NO2 and CO lifetimes, the different seasonality in concentrations should also be addressed. For example, is seasonality removed before computing the background CO? Also, in computing the

emission inventory ratios, are monthly emissions used when matching to data from a particular month, or do you apply annual averages?

Abstract. The abstract should state that NO2/CO is a proxy for combustion efficiency since combustion efficiency is a well-defined quantity: CO2/(CO2+CO). This would be better than calling it "burning efficiency", which is confusing since combustion and burning are the same.

Perhaps the title could be: "Quantifying NO2/CO using TROPOMI to characterize urban combustion"

Line 57 – should also reference Tang et al., 2019:

Tang, W., A. F. Arellano, B. Gaubert, K. Miyazaki, and H. M. Worden (2019), Satellite data reveal a common combustion emission pathway for major cities in China, Atmospheric Chemistry and Physics, 19(7), 4269–4288, doi:https://doi.org/10.5194/acp-19-4269-2019.

Line 85 – MOPITT also has a SWIR channel (or near IR) and the multispectral (TIR/NIR) product, with near-surface sensitivity over some land regions, was used in both Silva and Arellano, 2017 and Tang and Arellano, 2017.

---

## Author Comment (AC1) · 7 Jun 2020

This manuscript presents a novel application of satellite measurements of CO and NO2 to estimate regional-average burning ef?ciency for a number of large cities. The method is enabled by the capabilities of a relatively new satellite sensor and will likely interest many readers of ACP. With one major exception, the presented methods seem sound and the paper is generally well written. Thank you for your time and pointing out issues to improve the paper.

General Comments: To quantify the impact of this effect on Delta (XNO2)/Delta (XCO) ratios, the authors introduce the variable A_in?uence in Eq. 6. It is unclear how this

factor was derived or how it is calculated in practice; no derivation appears either in the main text or Appendices. Presumably, it somehow depends on the TROPOMI CO and NO2 averaging kernels, but these dependences are not presented. There is a paragraph on the effects of the differing averaging kernels at the bottom of p. 11 (lines 282-290), but this paragraph only adds to the confusion since nowhere does it actually refer to the variable A_influence:

Author Response: Ainfluence is the influence of the averaging kernel (A) on the model simulated NO2/CO column ratio. It is derived by calculating Delta (XNO2)/Delta (XCO) without and with the use of the averaging kernel as follows A_influence= ((Without A-with A))/(Without A ).100% and as mentioned in line 285. We have added Appendix C to further clarify how we derive the influence.

In the same paragraph, the authors report that "The CAMS simulated city enhancements averaged over June to August, 2018 did not compare well with TROPOMI for CO, possibly due to the coarse resolution of CAMS. Therefore, to calculate the averaging kernel impact, a few days were selected when CAMs CO and NO2 enhancements did compare relatively well with TROPOMI." This gives the impression that the authors' method of analysing the effects of the averaging kernel differences for CO and NO2 was based on a small number of 'cherry-picked' cases where the higher resolution of TROPOMI (compared to CAMS) was not an issue. Thus, it appears that the authors are probably underestimating the uncertainty of the averaging kernel-related error. I believe this entire issue requires more discussion and perhaps more analysis.

Author Response: The reason for using CAMS is to have realistic vertical profiles of NO2 and CO over cities, as those vertical profiles determine the impact that the averaging kernels will have on the ratio. What the reviewer calls "cherry picking", is actually a selection of cases that is representative of the conditions for which we use TROPOMI data in our analysis. This way the CAMS derived averaging kernel impact is expected to be representative of the impact on the actual TROPOMI data we use. The question is how variable this impact is, since averaging kernels generally do not vary much,

which has been tested for Tehran, Mexico City, Cairo, Riyadh, Lahore and Los Angeles. The results confirm that A_influence is about 10 -15 %. For Mexico City and Los Angeles, we used all the days from June-August, 2018 , see Table S2 (supplements).

Specific Comments 1. The actual lifetimes of CO and NO2 should be discussed somewhere, perhaps in the paragraph that begins on p. 3, l. 75 Author Response: The life time of CO and NO2 is discussed in the section 2.5 NO2/CO emission ratio, Line 196 to 198. There we also explain how we take in account of NO2 loss by OH, which leads to the short life time of NO2 during the noon time.

2. p. 4, l. 103. Rodgers (2000) does not specifically discuss this type of retrieval algorithm and is not really an appropriate reference. Mathematically, averaging kernels play a different role in optimal estimation-based methods (as described by Rodgers) and Tikhonov regularization. Author Response: The reference has been changed to Borsdorff(2018c). This reference is more appropriate as it explains how the CO total column is derived for TROPOMI

3. The chosen QA threshold values (0.75 for NO2 and 0.7 for CO) would seem to allow low-clouds for CO retrievals but not for NO2 retrievals. Are scenes with clouds excluded from this study because of the stricter QA threshold value for NO2? Clouds could have a significant impact on the TROPOMI CO column averaging kernels. Author Response: The application of SICOR algorithm on SCIAMACHY CO retrievals with low-level clouds increases the number of measurement with a limited impact on the retrieval quality (Borsdorff et al., 2018a). In addition, we selected pixels that had valid retrievals for both NO2 and CO. Therefore, CO and NO2 will be influenced similarly by the residual availability of clouds.

4. Conceptually, the Upwind Background and Plume Rotation methods seem to have much in common. The text in Sections 2.4.1 and 2.4.2 should somewhere discuss expected differences in the outcomes from these two methods. Are there any obvious pros and cons to each method?

Author Response: The Upwind Background method is used to calculate enhancement ratios based on single orbits, whereas in the Plume Rotation method column enhancement ratios are computed from CO and NO2 columns that are averaged for three months. The plume rotation method is used primarily in reference to what was done in the past (Pommier et al 2013), using MOPITT data that had to be averaged for city signals to be detectable. The use of TROPOMI data has the advantage that city enhancements are detected already in single satellite overpasses, which the Upwind Background method helps exploiting. The use of the two methods allows us to quantify the robustness of the emission ratio that we derived from TROPOMI.

5. For the Plume Rotation method, why use the fi̧rst quartile upwind and fourth quartile downwind concentrations (instead of simple averages for upwind and downwind regions)?

Author Response: We use the method of Pommier et al., 2013 to compare our own approach with, but decided to make it less vulnerable to outliers by taking quartiles following Silva and Arellano, 2017 instead of 5 max and 5 min data.

Technical Corrections (partial list) The numeral 2 should be subscripted in 'NO2' (in Abstract and elsewhere). Changed as suggested

Throughout the paper, 'mega cities' and 'mega-cities' should be replaced by 'megacities.' Changed as suggested

p. 2, l. 48. 'depends' should be 'depend' Changed as suggested

p. 2, l. 55. 'in respect' should be 'with respect' Changed as suggested

p. 2, l. 66. 'precursor' should be capitalized Changed as suggested

p. 3, l. 76. 'source' should be 'sources' Changed as suggested

p. 8, l. 229. 'over passes' should be 'overpasses' Changed as suggested

p. 8, l., 232. 'life time' should be 'lifetime' Changed as suggested

p. 9, l. 244. missing Delta symbols before XNO2 and XCO Changed as suggested

Please also note the supplement to this comment:
https://www.atmos-chem-phys-discuss.net/acp-2019-1112/acp-2019-1112-AC1-
supplement.pdf

---

## Author Comment (AC2) · 7 Jun 2020

**#Reveiwer 2:  Quantifying burning efficiency in Megacities using NO2 /CO ratio from the Tropospheric Monitoring Instrument (TROPOMI)"**

The paper presents NOx/CO emission ratios as derived from satellite observations of NO2 and CO. It demonstrates the high potential of TROPOMI for atmospheric research. The paper is generally well written. However, the details of the method lack some details, and the robustness of the results is hard to evaluate with the given information. Thus, before publication, major revisions are necessary. The method description has to be extended. In particular, the two approaches have to be illustrated for real data rather than just for a schematic plot. In addition, the definitions of background and upwind areas have to be made consistent for all cities. If this is not possible add a discussion of uncertainties due to the a-priori settings. Error bars only account for statistical day-to-day variations, but ignore the impact of a-priori settings and the CER procedure.

*Author Response:*

*Thank you for your time and suggestions, particularly concerning details of the method that we used, which helped to improve the manuscript.*

**Detailed comments:**
**- Line 117: Is this bias in NO2 accounted for in your study? How would the results change if NO2 would be scaled up accordingly?**

*Author Response:*

*The bias in S5P NO2 retrieval has been assessed for European cities. However, since we don't know yet how representative this estimate is for the cities that we study, it was decided to account for the impact of the bias as an additional the source of uncertainty of 25% of the TROPOMI inferred NO2/CO ratio (see Table S3).*

*In case, if 25 % bias would apply for all the cities, TROPOMI derived emission ratio increase by factor of 1.25. TROPOMI derived emission ratio for Mexico City remains close to MACCITY (within 5%) and EDGAR (within 25%). For Tehran, Cairo, Lahore and Riyadh MACCity emission ratio is lower by 50 % in contrast to TROPOMI derived ratio.  EDGAR emission ratio is close to TROPOMI derived ratio for Riyadh and Lahore (within 25 %). For Los Angeles, TROPOMI derived emission ratio is lower by factor 2 and 3 than EDGAR and MACCity ratio.*

[Figure]

**Figure 1.** Comparison of TROPOMI-derived Upwind Background Corrected emission ratio shown in blue shades, to corresponding emission ratios from the EDGAR (red) and MACCity (yellow) emission inventories for six *megacities*. Error bars represent 1σ uncertainties calculated using boot strapping method.

**- Table 1 settings: I am puzzled by the different definitions for different cities, especially as Line 146 states that the settings are "not critical". So why are they different at all? Do you need to tune the area definitions in order to get the right results??? How do the results look like if a consistent setup is chosen for all cities? Why is the upwind area for Riyadh different for dlat and dlon by a factor of 30?**
*Author Response:*
*The sentence " Every city has a different size and different neighboring CO and NO2 emission sources and therefore the appropriate choice of radii for the background and outskirt areas varies between cities (detail explanation in Supplements Section 1). This is important mostly to have a significant signal from city emissions in CO and NO2. However, since the same regional definition is used for NO2 and CO, the enhancement ratio is not so sensitive to the details of the region selection." is added in the line 154 to 158.*

*We performed the sensitivity analysis where background and outskirt radius is increased by 10km for four times results in the ratio change by 20 % for Riyadh and < 10 % for all the cities (see Figure S21).*

*There was a typing error for dlat and dlon of Riyadh. To maintain the consistency now, I am using 1.0˚, 1.0˚ dlat and dlon for all the cities.*

**Page 6:**
**the methods are explained in plain words, but not illustrated for real data. So Fig. S2 should be moved to the main text, and the background/upwind regions etc. should be marked in this plot. In addition, the ERA wind vector should be added. The rotated patterns and the percentiles used for the second approach should be provided in a separate figure.**
*Author Response:*

*Figure1 is added in the paper, to illustrate the upwind background method for real data as suggested by the reviewer. For the plume rotation method, see the Figure S8.*

[Figure]

**Figure 2.** ERA interim average wind speed and direction from surface to 200m at the time TROPOMI overpasses (left ) and TROPOMI derived CO total column over Mexico City (right) for 5th of June, 2018. The black star represents the centre of the city. In the right panel, the white circle is the background area for Mexico City and the blue section represents the upwind background area that we selected depending upon the wind direction in the core city area. P0,P1,P2 and P3 are the points where north, east, west and south wind directions intersects at the inner rim of the background area.

[Figure]

**Figure S8.** TROPOMI derived XCO a) averaged over June-August 2018 b) plume rotated over Mexico, c) Upwind and d) downwind region. The white star in top panel is the centre of Mexico City. The white lines in panel (b) represent the 20x100km² area to determine the column enhancement in the city. The area to the North and South of the city centre is upwind and

downwind region respectively. The red line in panel c) and d) represents the 25 percentile and 75 percentile respectively. CO retrievals are gridded in 0.1˚x0.1˚

**both methods compare columns "upwind" and "downwind" of the investigated cities. This approach requires that there \*is\* transport taking place. Wind speeds for Mexico City are quite low, as can also be seen in Fig. S2. So did you consider a minimum threshold for the wind speed? I expect that it would help to remove inconclusive days.**

*Author Response:*
*In this study we do not use a minimum threshold for the wind speed, because depending on the choice which would inevitably be quite arbitrary and a substantial amount of useful data may be lost. However, as explained before, by studying the ratio between NO2 and CO we are less sensitive to transport issues. Most important is to have a method the quantifies the difference between city and background in a way that is consistent between CO and NO2.*

**- Line 157: with Eqs 2&3, daily ratios are calculated. But how is the total ratio (shown in Fig. 3) derived? Is it the mean of all daily ratios? This is by definition different from the second approach, where first CO and NO2 are averaged and then the ratio of means is calculated. Thus, also for approach 1, the ratio of means should be taken.**

*Author Response:*
*In the upwind background method, the mean is taken of daily ratios. This method is favorable over the use of mean NO2 and CO in the plume rotation method, because it accounts for temporal correlations between NO2 and CO. However, for the plume rotation method we choose to stay consistent with Pommier et al (2013), which averages first since it is important when using noisy MOPITT data. The implication is that there is the inconsistency that the reviewer mentions. However, by including it in the comparison, we also implicitly test the robustness of the TROPOMI derived ratio to this methodological difference.*

**- Line 196: What emission database is used by CAMS? EDGAR? MACCity ? Or something else ? How far does this affect the following interpretation and discussion of CAMS OH? How do CAMS spatial patterns of CO and NO2 compare to TROPOMI? Please provide a Figure in the Supplement. Is TROPOMI CO and/or NO2 assimilated in CAMS?**

*Author Response:*
*CAMS is using MACCity for the anthropogenic emission. CAMS OH concentration depends upon the various chemical schemes used for the simulation rather than emission inventories (V. Huijen et al., 2019). The spatial pattern of NO2 of CAMS and TROPOMI are in good agreement for the six different cities (see Section 5 in Supplements Fig S11 to Fig S16). However, the spatial distribution of CO shows differences for Tehran, Cairo, Riyadh and Lahore (see Section 5 in Supplements Fig S13 to Fig S16). CAMS is using MOPPIT for CO and for $NO_2$, SCIAMACHY, GOME2 and OMI data are assimilated. TROPOMI CO and NO2 are not yet included in CAMS.*

[Figure]

**Figure S12.** Collocated XCO (top) and XNO2 (bottom) averaged for June-August, 2018 over Mexico City and derived from TROPOMI (left) and CAMS (right). The white star represents the centre of the city. The enhancement of XCO and XNO2 in TROPOMI and CAMS collocates.CO and NO2 retrievals are gridded at 0.1˚x0.1˚ resolution.

[Figure]

**Figure S13.** Same as FigS10 but over Los Angeles

[Figure]

**Figure S14.** Collocated XCO (top) and XNO2 (bottom) averaged for June-August, 2018 over Tehran and derived from TROPOMI (left) and CAMS (right). The white star represents the centre of the city. The enhancement of CAMS XCO does not collocate with the TROPOMI XCO at the centre of city whereas NO2 enhancement collocates with each other. CO and NO2 retrievals are gridded at 0.1˚x0.1˚ resolution.

[Figure]

**Figure S15.** Same as Figure S12 but over Cairo

[Figure]

**Figure S16.** Same as Figure S13 but over Riyadh

[Figure]

**Figure S17.** Same as Figure S13 but over Lahore

**2.6.1: The bootstrapping approach evaluates the statistical uncertainty for the results with the chosen approach. But on top, there are also other uncertainties, like systematic effects introduced by the definition of radii etc. In particular the uncertainties of the wind direction and wind speed have to be discussed as well.**

*Author Response:*

*As suggested by the reviewer, we tested the sensitivity to the wind speed and direction by choosing different heights (i.e. 200m to 1000m), resulting in differences < 10 % for all the cities (see Figure S19, S20).*

[Figure]

**Figure S19.** TROPOMI derived Upwind Background Corrected Emission Ratio for six megacities using average wind speed and direction calculated from surface till 200m to 1000m. The error bar represents 1σ uncertainties calculated using boot strapping.

[Figure]

**Figure S20.** Same as Fig S11 but the emission ratio is Plume Rotation Corrected Emission ratio

*We performed the sensitivity analysis where background and outskirt radius is increased by 10km for four times results in the uncertainty of 7-20 % in TROPOMI column enhancement ratio for six megacities (see Figure S21).*

[Figure]

**Figure S21.** Upwind Background Corrected Emission ratio derived for six megacities using four different background and outskirt radius. For the initial step the outskirt and background radius for Tehran: 180 km and 190 km, Mexico City: 170km and 180km, Cairo: 135km and 145 km, Riyadh: 100km and 110 km, Lahore: 100km and 110km and Los Angeles: 200 km and 210 km. In every step the background and outskirt radius is increased by 10km.During this process dlat and dlon is 1.0˚,1.0 ˚.

**Figure 3: Please estimate the uncertainties of NO2 lifetime and AK correction and provide error bars for the CER results as well. I expect that these uncertainties are far higher (and thus more relevant) than the purely statistical bootstrap uncertainties.**

*Author Response:*
*Table S3 shows the different sources of uncertainty (i.e. NO2 lifetime, Ak correction, wind direction and their contribution to the TROPOMI derived emission ratio). Error bars indicating the 1σ uncertainties in TROPOMI derived emission ratios are added in Figure 4. The following sentence is added to line 347 to 355 explaining the different sources of uncertainty and their effect on TROPOMI derived emission ratios:*

*"We calculated the wind direction and wind speed at different height i.e. 200m to 1000m and the ratio changes<10 % for all the cities(FigS19 and S20). The initial uncertainty for CAMS OH was ±50 % (V. Huijen et al., 2019). The bootstrapping method show that the concentration of OH varies from 8.0 − 15 % for six different megacities resulting similar uncertainty to the TROPOMI derived emission ratio. If the CAMS overestimate OH concentration systematically, the TROPOMI derived emission ratio will decrease. To estimate the effect of predefined areas as background, we simultaneously increase the outskirt and background radius by 10 km for all the cities for four times. The effect is about 20 % for Riyadh whereas for other cities, the effect is < 12 % (Fig S21). S5P TROPOMI NO2 retrievals have the largest contribution for the total uncertainty on satellite derived emission ratio. The wind direction and speed, boundary layer OH concentration, Ainfluence correction and the predefined background setting contributes the negligible uncertainty on the TROPOMI derived emission ratio. The total*

*uncertainty calculated using error propagation method for TROPOMI derived emission ratio ranges from 27 to 35 % and the detail is provided in Table S3."*

**Table S3. Sources of uncertainties for TROPOMI derived emission ratio. The total uncertainty is derived by the error propagation.**

| City | S5P TROPOMI $NO_2$ uncertainty (%) | Wind direction and Wind speed (%) | Boundary layer OH concentration (%) | Predefined background area (%) | $A_{influence}$ correction (%) | Total effect on ER (%) |
|---|---|---|---|---|---|---|
| Tehran | 25 | 1.5 | 8.4 | 10 | 1.2 | ±28.2 |
| Mexico City | 25 | 1.5 | 10 | 7 | 2.7 | ±27.9 |
| Cairo | 25 | 2.6 | 8.4 | 10 | 4.4 | ±28.6 |
| Riyadh | 25 | 2.0 | 12.5 | 20 | 4.1 | ±34.6 |
| Lahore | 25 | 6.5 | 15.0 | 12.0 | 0.4 | ±32.19 |
| Los Angeles | 25 | 4.0 | 8.3 | 12.5 | 4.2 | ±29.7 |

**- 3.2ff: Please check the discussion and conclusions (a) for NO2 probably being biased low and (b) according to the quality of CAMS emissions and the agreement between TROPOMI and CAMS**

*Author Response:*

*Discussion section: Lines 346-348 discuss the importance of the bias in NO2. The sentence is as follows:" Additionally, TROPOMI underestimates $NO_2$ column by 7 % to 29.7 % relative to MAX-DOAS ground based measurement in European cities (Lambert, et al., 2019). However, since we don't know yet how representative this estimate is for the cities that we study so, the impact of the bias is accounted as an additional the source of uncertainty of 25% of the TROPOMI inferred NO2/CO ratio (see Table S3)".*

*Lines 318-320 describe the agreement between TROPOMI and CAMS over Mexico City and Los Angeles. The sentence is as follows:*
*"CAMS derived enhancement ratio for Mexico City differs by 5 % compared to UB and PR but for Los Angeles the ratio (0.094) is higher by 75% compared to UB and PR (0.034)."*

*Conclusion Section:*
*-Line 428 to 430 explains about the uncertainties and NO2 being biased low. The sentence is as follow:*
*"The total uncertainty on TROPOMI derived emission ratio ranges from 27 to 35 %. The bias in S5P TROPOMI NO2 retrievals accounts for the major contribution for the uncertainties in the TROPOMI derived emission ratio".*

*Line 434 to 435 explains about the agreement between TROPOMI and CAMS over Mexico City. The sentence is "CAMS derived enhancement ratio over Mexico City differs by 5 % compared to UB and*

*PR". Line 438 to 439 explains about the disagreement between TROPOMI and CAMS over Los Angeles. The sentence is as follow:*

*"CAMS derived enhancement ratio for Los Angeles is higher by 75 % in contrast to UB and PR "*

**Minor comments:**
**- Lines 40-42: The references stating the high uncertainty of Chinese emissions are from a time period where development in China was vastly increasing. Meanwhile, NOx emissions have been reduced, and the awareness of air pollution has increased in China. I would thus assume that these high uncertainties do not hold any longer.**
*Author Response:*

*- The uncertainty in emission estimates of 2005 to 2008 is added in the sentence.*

**- Lines 60-61 Please provide refs to SCIAMACHY (Bovensmann) and TROPOMI (Veefkind).**
*- Changed as suggested*

**- Line 70: Should be NOx emission.**
*- Changed as suggested*

**- Line 75: Transport disperses NO2 and CO similarly, but the lifetime of NO2 is far shorter! See the different plume extents shown in Fig. S2.**
*Author Response:*
*The reviewer is right that the plume extents are different due to the difference on lifetime, which is the reason why we focus on the core city area rather than the full extent of the plume.*

**- Line 117: Avoid misreading as "the bias is low", e.g. "NO2 is biased low by about**

*- Changed as suggested*

**- Table 1 lat/lon: Please provide consistent number of digits for lat/lon. .01_ should be accurate enough.**
*- changed as suggested*

**- Fig. S1: I don't understand why there is a need for separating 4 different wind directions in the formalism; rotation matrix should work the same for all four cases!?**

*Author Response:*

*-changed as suggested and see Figure S7.*

[Figure]

**Figure S7.** Schematic representation of the procedure used for selecting the upwind background. The centre of the city is represented by red star. The city, outskirt and background radii (see Table 1) are used to divide the city into three parts i.e. the core city (red circle), outskirt (white circle) and background region (blue circle), respectively. Step 1.Selection of radius R1, the mean of outskirt and background radii. Select the points P0, P1, P2, P3 where the north, east, south and west wind directions (**θ**) intersect at the outer rim of the dashed circle with radius R1. The black arrow symbolises an average wind direction over the core city region. Step 2.The rotation of P0 with **θ** in reference to the city centre and generate the new point. Step 3.Select the fraction of the upwind region. The Δlat and Δlon is provided in Table1.

---

## Author Comment (AC3) · 7 Jun 2020

This study presents new results from TROPOMI for NO2/CO emission factors that provide information about combustion efficiency on urban scales. This is an important result for understanding how well these emissions are represented in standard inventories with subsequent impacts for air quality and climate model predictions. I recommend publication after the comments from 2 other referees and some minor issues from me are addressed.

*Author Response:*

*Thank you for your time and comments to improve this paper.*

Following the comment of Ref.#1 in addressing the different NO2 and CO lifetimes, the different seasonality in concentrations should also be addressed. For example, is seasonality removed before computing the background CO? Also, in computing emission inventory ratios, are monthly emissions used when matching to data from a particular month, or do you apply annual averages?

*Author Response:*

*To address this point, we switched to monthly emission, using EDGAR v4.3.2 2010 and MACCity 2018. The seasonal correction factor is quantified using EDGAR v4.3.2 2010 since monthly data for EDGAR 2012 is not available (see Fig S18). June to August (JJA) EDGAR 2012 ratio reduces by < 12% for Tehran, Cairo, Riyadh and Mexico City in contrast to annual average inventory derived ratio. However, in JJA MACCity ratio increase by 27.0% Tehran, 10 % for Mexico City, 50 % Cairo and 71 % for Lahore (see Fig 4). The JJA MACCity ratio is close to UBCER and PECER (within 10 %) for all the cities except Los Angeles. EDGAR and MACCity do not agree on the seasonal effect on the emission and comparison of seasonal ratio might result uncertainty in inventory derived ratio. The sentence is added in line 335 to 345.*

[Figure]

**Figure S18. Comparison of EDGAR v4.3.2, 2010 and MACCity 2018 derived emission ratio using annual average emission (dark solid color ) and June to August averaged emission (faded color ) to the TROPOMI derived emission ratio ( blue shades)**

[Figure]

**Figure 1.** Comparison of TROPOMI-derived $\Delta NO_2/\Delta CO$ enhancement ratios, calculated using different methods shown in blue shades, to corresponding emission ratios from the EDGAR (red shades) and MACCity (yellow shades) emission inventories for six *megacities*. The dark solid shades for emission inventories represent the annual average inventory derived ratio whereas faded shades represents the June to August average inventory derived ratio. Error bars represent 1σ uncertainties calculated using boot strapping (upwind background) and error propagation (plume rotation method). The upwind background corrected emission ratio (UBCER) and Plume rotation corrected emission ratio (PRCER) account for the impact of photochemical $NO_2$ removal and the averaging kernel.

**Abstract. The abstract should state that NO2/CO is a proxy for combustion efficiency since combustion efficiency is a well-defined quantity: CO2/(CO2+CO). This would be better than calling it "burning efficiency", which is confusing since combustion and burning are the same.**

*Author Response:*

*In the introduction section Line 74 to 77: "We use the ratio of the TROPOMI retrieved tropospheric column of $NO_2$ and the total column of CO, which is formally not equivalent to combustion efficiency but can nevertheless serve as a useful proxy (Silva & Arellano, 2017; W. Tang & Arellano, 2017). The reason for this is that $NO_x$ emission increases with combustion temperature, which is high during efficient combustion. In contrast, CO is a product of incomplete combustion, and is produced when combustion efficiency is low (Flagan & Seinfeld, 1988).The combination of these effects makes the $NO_2/CO$ ratio highly sensitive to combustion efficiency" make the things clear about the combustion and burning efficiency.*

*However I have added the sentence in the abstract to avoid the confusion in line 15 to 17. The sentence is as follows:" NOx (NO+NO2) emission increases during the efficient combustion whereas*

*incomplete combustion results to higher CO emission. Therefore, NO2/CO is a good proxy for combustion efficiency"*

**Perhaps the title could be: "Quantifying NO2/CO using TROPOMI to characterize urban combustion"**

*Author Response:*

*Thank you for the suggestion but we are not formally quantifying combustion efficiency. However, we deliberately do not use the term 'combustion efficiency'. Therefore we choose to keep the old formulation of the title and explain carefully in the introduction section what we mean by burning efficiency.*

**Line 57 – should also reference Tang et al., 2019:**

*-changed as suggested*

**Line 85 – MOPITT also has a SWIR channel (or near IR) and the multispectral (TIR/NIR) product, with near-surface sensitivity over some land regions, was used in both Silva and Arellano, 2017 and Tang and Arellano, 2017.**

*-changed as suggested*

---

## Author Comment (AC5) · 3 Jul 2020

This manuscript presents a novel application of satellite measurements of CO and NO2 to estimate regional-average burning efficiency for a number of large cities. The method is enabled by the capabilities of a relatively new satellite sensor and will likely interest many readers of ACP. With one major exception, the presented methods seem sound and the paper is generally well written.

Author Response:

Thank you for your time and pointing out issues to improve the paper.

[Figure]

General Comments:

1. To quantify the impact of this effect on Delta (XNO2)/Delta (XCO) ratios, the authors introduce the variable A_influence in Eq. 6. It is unclear how this factor was derived or how it is calculated in practice; no derivation appears either in the main text or Appendices. Presumably, it somehow depends on the TROPOMI CO and NO2 averaging kernels, but these dependences are not presented. There is a paragraph on the effects of the differing averaging kernels at the bottom of p. 11 (lines 282-290), but this paragraph only adds to the confusion since nowhere does it actually refer to the variable A_influence.

Author Response:

A_influence is the influence of the averaging kernel (A) on the TROPOMI observed NO2/CO column ratio. It is derived by calculating Delta (XNO2)/Delta (XCO) without and with the use of the averaging kernel applied the vertical profiles of NO2 and CO from the CAMS reanalysis, as follows A_influence= ((Without A-with A))/(Without A ).100% and as mentioned in line 305. We have added Appendix C to further clarify how we derive the influence.

2. In the same paragraph, the authors report that "The CAMS simulated city enhancements averaged over June to August, 2018 did not compare well with TROPOMI for CO, possibly due to the coarse resolution of CAMS. Therefore, to calculate the averaging kernel impact, a few days were selected when CAMs CO and NO2 enhancements did compare relatively well with TROPOMI." This gives the impression that the authors' method of analysing the effects of the averaging kernel differences for CO and NO2 was based on a small number of 'cherry-picked' cases where the higher resolution of TROPOMI (compared to CAMS) was not an issue. Thus, it appears that the authors are probably underestimating the uncertainty of the averaging kernel-related error. I believe this entire issue requires more discussion and perhaps more analysis.

Author Response:

The reason for using CAMS is to have realistic vertical profiles of NO2 and CO over cities, as those vertical profiles determine the impact that the averaging kernels will have on the ratio. What the reviewer calls "cherry picking", is actually a selection of cases that is representative of the conditions for which we use TROPOMI data in our analysis. This way the CAMS derived averaging kernel impact is expected to be representative of the impact on the actual TROPOMI data we use. The question is how variable this impact is, since averaging kernels generally do not vary much, which has been tested for Tehran, Mexico City, Cairo, Riyadh, Lahore and Los Angeles. The results confirm that $A\_influence$ is about 10 -15 %. For Mexico City and Los Angeles, we used all the days from June-August, 2018 , see Table S2.

Specific Comments

1. The actual lifetimes of CO and NO2 should be discussed somewhere, perhaps in the paragraph that begins on p. 3, l. 75

Author Response:

The lifetime of CO and NO2 is discussed in the section 2.5 NO2/CO emission ratio, Line 200 to 205. There we also explain how we take the NO2 loss by OH into account, which leads to a short lifetime of NO2 at the local overpass time of TROPOMI. The much slower photochemical turnover of CO can be neglected on the temporal and spatial scale of our analysis.

2. p. 4, l. 103. Rodgers (2000) does not specifically discuss this type of retrieval algorithm and is not really an appropriate reference. Mathematically, averaging kernels play a different role in optimal estimation-based methods (as described by Rodgers) and Tikhonov regularization.

Author Response:

The reference has been changed to Borsdorff(2018c). This reference is more appropriate as it explains how the CO total column is derived for TROPOMI.

3. The chosen QA threshold values (0.75 for NO2 and 0.7 for CO) would seem to allow low-clouds for CO retrievals but not for NO2 retrievals. Are scenes with clouds excluded from this study because of the stricter QA threshold value for NO2? Clouds could have a significant impact on the TROPOMI CO column averaging kernels.

Author Response:

The application of SICOR algorithm to SCIAMACHY CO retrievals with low-level clouds increases the number of measurement with a limited impact on the retrieval quality (Borsdorff et al., 2018a). In addition, we selected pixels that had valid retrievals for both NO2 and CO. Therefore, CO and NO2 will be influenced similarly by the residual availability of clouds.

4. Conceptually, the Upwind Background and Plume Rotation methods seem to have much in common. The text in Sections 2.4.1 and 2.4.2 should somewhere discuss expected differences in the outcomes from these two methods. Are there any obvious pros and cons to each method?

Author Response:

The Upwind Background method is used to calculate enhancement ratios based on single orbits, whereas in the Plume Rotation method column enhancement ratios are computed from CO and NO2 columns that are averaged for three months. The plume rotation method is used primarily in reference to what was done in the past (Pommier et al 2013), using MOPITT data that had to be averaged for city signals to be detectable. The use of TROPOMI data has the advantage that city enhancements are detected already in single satellite overpasses, which the Upwind Background method helps exploiting. The use of the two methods allows us to quantify the robustness of the emission ratio that we derived from TROPOMI.

5. For the Plume Rotation method, why use the first quartile upwind and fourth quartile downwind concentrations (instead of simple averages for upwind and downwind

regions)?

Author Response:

We use the method of Pommier et al., 2013 to compare our own approach with, but decided to make it less vulnerable to outliers by taking quartiles following Silva and Arellano, 2017 instead of 5 max and 5 min data.

Technical Corrections (partial list)

The numeral 2 should be subscripted in 'NO2' (in Abstract and elsewhere). Changed as suggested

Throughout the paper, 'mega cities' and 'mega-cities' should be replaced by 'megacities.' Changed as suggested

p. 2, l. 48. 'depends' should be 'depend' Changed as suggested

p. 2, l. 55. 'in respect' should be 'with respect' Changed as suggested

p. 2, l. 66. 'precursor' should be capitalized Changed as suggested

p. 3, l. 76. 'source' should be 'sources' Changed as suggested

p. 8, l. 229. 'over passes' should be 'overpasses' Changed as suggested

p. 8, l., 232. 'life time' should be 'lifetime' Changed as suggested

Please also note the supplement to this comment:
https://www.atmos-chem-phys-discuss.net/acp-2019-1112/acp-2019-1112-AC5-supplement.pdf

---

## Author Comment (AC6) · 3 Jul 2020

Please note that for the Figures look at the supplementary file attached below.

The paper presents NOx/CO emission ratios as derived from satellite observations of NO2 and CO. It demonstrates the high potential of TROPOMI for atmospheric research. The paper is generally well written. However, the details of the method lack some details, and the robustness of the results is hard to evaluate with the given information. Thus, before publication, major revisions are necessary. The method description has to be extended. In particular, the two approaches have to be illustrated for real data rather than just for a schematic plot. In addition, the definitions of background and

upwind areas have to be made consistent for all cities. If this is not possible add a discussion of uncertainties due to the a-priori settings. Error bars only account for statistical day-to-day variations, but ignore the impact of a-priori settings and the CER procedure.

Author Response:

Thank you for your time and suggestions, particularly concerning details of the method that we used, which helped to improve the manuscript.

Detailed comments:

- Line 117: Is this bias in NO2 accounted for in your study? How would the results change if NO2 would be scaled up accordingly?

Author Response:

The bias in S5P NO2 retrieval has been assessed for European cities. However, since we don't know yet how representative this estimate is for the cities that we study, it was decided to account for the impact of the bias as an additional the source of uncertainty of 30% of the TROPOMI inferred NO2/CO ratio (see Table S3). If the 30 % bias would apply to all the cities, then the TROPOMI derived emission ratios increase by factor of 1.30. In that case, TROPOMI-derived ratios remain in line with the emission inventories but the agreement would shift between EDGAR and MACCity depending on the city, for example in favor of EDGAR in case of Cairo and Riyadh. For Los Angeles, the difference between TROPOMI and the inventories would decrease, but remain significantly lower for TROPOMI.

Table 1 settings: I am puzzled by the different definitions for different cities, especially as Line 146 states that the settings are "not critical". So why are they different at all? Do you need to tune the area definitions in order to get the right results??? How do the results look like if a consistent setup is chosen for all cities? Why is the upwind area for Riyadh different for dlat and dlon by a factor of 30?

Author Response:

The following sentence was added (line 154-163): "Every city has a different size and different neighboring CO and NO2 emission sources and therefore the appropriate choice of radii for the background and outskirt areas varies between cities (see Supplements Section 1 for details). Since the same regional definition is used for NO2 and CO, the enhancement ratio is not sensitive to the details of the region selection. Most important for the choice of radii is to catch the local enhancement in CO and NO2 to its full extend, to optimize the signal over noise and thereby the detection limit for urban emissions." We performed a sensitivity test in which the background and outskirt radii were increased by 10km in four steps resulting in the ratio changes by < 15 % for all the cities (see Figure S20). There was a typing error for $\Delta$lat and $\Delta$lon of Riyadh. To maintain the consistency now, I am using $1.0°$, $1.0°$ $\Delta$lat and $\Delta$lon for all the cities.

Page 6:

the methods are explained in plain words, but not illustrated for real data. So Fig. S2 should be moved to the main text, and the background/upwind regions etc. should be marked in this plot. In addition, the ERA wind vector should be added. The rotated patterns and the percentiles used for the second approach should be provided in a separate figure.

Author Response:

Figure1 is added in the paper, to illustrate the upwind background method for real data as suggested by the reviewer. For the plume rotation method, see the Figure S8.

both methods compare columns "upwind" and "downwind" of the investigated cities. This approach requires that there *is* transport taking place. Wind speeds for Mexico City are quite low, as can alsobe seen in Fig. S2. So did you consider a minimum threshold for the wind speed? I expect that it would help to remove inconclusive days.

Author Response:

In this study we do not use a minimum threshold for the wind speed, because depending on the choice which would inevitably be quite arbitrary and a substantial amount of useful data may be lost. However, as explained before, by studying the ratio between NO2 and CO we are less sensitive to transport issues. Most important is to have a method the quantifies the difference between city and background in a way that is consistent between CO and NO2.

- Line 157: with Eqs 2&3, daily ratios are calculated. But how is the total ratio (shown in Fig. 3) derived? Is it the mean of all daily ratios? This is by definition different from the second approach, where first CO and NO2 are averaged and then the ratio of means is calculated. Thus, also for approach 1, the ratio of means should be taken.

Author Response:

In the upwind background method, the mean is taken of daily ratios. This method is favorable over the use of mean NO2 and CO in the plume rotation method, because it accounts for temporal correlations between NO2 and CO. However, for the plume rotation method we choose to stay consistent with Pommier et al (2013), which averages first since it is important when using noisy MOPITT data. The implication is that there is the inconsistency that the reviewer mentions. However, by including it in the comparison, we also implicitly test the robustness of the TROPOMI derived ratio to this methodological difference.

- Line 196: What emission database is used by CAMS? EDGAR? MACCity ? Or something else ? How far does this affect the following interpretation and discussion of CAMS OH? How do CAMS spatial patterns of CO and NO2 compare to TROPOMI? Please provide a Figure in the Supplement. Is TROPOMI CO and/or NO2 assimilated in CAMS?

Author Response:

CAMS uses MACCity for the anthropogenic emission. CAMS OH depends upon the

chemical scheme used for the simulation in addition to the emission inventories (V. Huijen et al., 2019). We do not discuss or interpret CAMS OH. We only use it to derive an estimate of the NO2 lifetime, which is valid within the uncertainty bounds of the reanalysis. The spatial patterns of CAMS and TROPOMI NO2 are in reasonably good agreement for the six different cities (see Section 5 in Supplements Fig S12 to Fig S17). However, larger differences are found for CO over Tehran, Cairo, Riyadh and Lahore (see Section 5 in Supplements Fig S14 to Fig S17). CAMS is using MOPPIT for CO and for NO2, SCIAMACHY, GOME2 and OMI data are assimilated. TROPOMI CO and NO2 are not yet included in CAMS.

2.6.1: The bootstrapping approach evaluates the statistical uncertainty for the results with the chosen approach. But on top, there are also other uncertainties, like systematic effects introduced by the definition of radii etc. In particular the uncertainties of the wind direction and wind speed have to be discussed as well.

Author Response:

As suggested by the reviewer, we tested the sensitivity to the wind speed and direction by choosing different heights (i.e. 200m to 1000m), resulting in differences $\leq 10$ % for all the cities (see Figure S18, S19).

We also performed a sensitivity analysis in which the background and outskirt radii were increased by 10km in four steps resulting in an uncertainty of 7 to 15 % in TROPOMI column enhancement ratios for the six megacities (see Figure S20).

Figure 3: Please estimate the uncertainties of NO2 lifetime and AK correction and provide error bars for the CER results as well. I expect that these uncertainties are far higher (and thus more relevant) than the purely statistical bootstrap uncertainties.

Author Response:

Table S3 shows the different sources of uncertainty (i.e. NO2 lifetime, Ak correction, wind direction and their contribution to the TROPOMI derived emission ratio). Error

bars indicating $1\sigma$ uncertainties in TROPOMI derived emission ratios are added in Figure 4. The following sentence is added to line 355 to 360 explaining the different sources of uncertainty and their effect on TROPOMI derived emission ratios:

"Additionally, TROPOMI underestimates the NO2 column by 7 % to 29.7 % relative to MAX-DOAS ground based measurement in European cities (Lambert, et al., 2019). However, since we don't know yet how representative this estimate is for the cities that we study, the impact of this bias has been accounted for as an additional source of uncertainty of 30% in the TROPOMI inferred NO2/CO ratio (see Table S3). Compared to this number, other sources of uncertainty such as in the wind direction and speed (FigS18 and S19), boundary layer OH concentration, Ainfluence correction and the pre-defined background setting (Fig S20) make only small contributions to the TROPOMI derived emission ratio. The total uncertainty in the TROPOMI derived emission ratio is calculated using error propagation (see Table S3) and ranges between 33 to 35.6%."

- 3.2ff: Please check the discussion and conclusions (a) for NO2 probably being biased low and (b) according to the quality of CAMS emissions and the agreement between TROPOMI and CAMS

Author Response:

Discussion section: Lines 355-360 discuss the importance of the bias in NO2. The sentence is as follows:" Additionally, TROPOMI underestimates the NO2 column by 7 % to 29.7 % relative to MAX-DOAS ground based measurement in European cities (Lambert, et al., 2019). However, since we don't know yet how representative this estimate is for the cities that we study, the impact of this bias has been accounted for as an additional source of uncertainty of 30% in the TROPOMI inferred NO2/CO ratio (see Table S3)."

CAMS use multiple datasets i.e. MACCity emission inventory, SCIAMACHY, GOME2 and OMI data for CO and NO2. We don't know the influence of each of these datasets on the result. Therefore, we decided not to include CAMS in the comparison to

[Figure]

TROPOMI derived emission ratios.

Conclusion Section:

- Lines 435 to 436 explain about the uncertainties due to NO2 being biased low:" The total uncertainty on TROPOMI derived emission ratio ranges from 33 to 35.6%.The bias in S5P TROPOMI NO2 retrievals has the most important contribution to the uncertainty in the TROPOMI derived emission ratio".

Minor comments:

- Lines 40-42: The references stating the high uncertainty of Chinese emissions are from a time period where development in China was vastly increasing. Meanwhile, NOx emissions have been reduced, and the awareness of air pollution has increased in China. I would thus assume that these high uncertainties do not hold any longer.

Author Response:

- We have modified the sentence to clarify which years the uncertainty estimate referred to.

- Lines 60-61 Please provide refs to SCIAMACHY (Bovensmann) and TROPOMI (Veefkind).

- Changed as suggested

- Line 70: Should be NOx emission.

- Changed as suggested

- Line 75: Transport disperses NO2 and CO similarly, but the lifetime of NO2 is far shorter! See the different plume extents shown in Fig. S2.

Author Response:

The reviewer is right that the plume extents are different due to the difference on lifetime, which is the reason why we focus on the core city area rather than the full extent

of the plume.

- Line 117: Avoid misreading as "the bias is low", e.g. "NO2 is biased low by about

- Changed as suggested

- Table 1 lat/lon: Please provide consistent number of digits for lat/lon. .01_ should be accurate enough.

- changed as suggested

- Fig. S1: I don't understand why there is a need for separating 4 different wind directions in the formalism; rotation matrix should work the same for all four cases!?

Author Response:

-changed as suggested and see Figure S7.

Please also note the supplement to this comment:
https://www.atmos-chem-phys-discuss.net/acp-2019-1112/acp-2019-1112-AC6-supplement.pdf

———————————————

[Figure]

**Supplement:**

**#Reviewer 2:**  **Quantifying burning efficiency in Megacities using NO2 /CO ratio from the Tropospheric Monitoring Instrument (TROPOMI)"**

The paper presents NOx/CO emission ratios as derived from satellite observations of NO2 and CO. It demonstrates the high potential of TROPOMI for atmospheric research. The paper is generally well written. However, the details of the method lack some details, and the robustness of the results is hard to evaluate with the given information. Thus, before publication, major revisions are necessary. The method description has to be extended. In particular, the two approaches have to be illustrated for real data rather than just for a schematic plot. In addition, the definitions of background and upwind areas have to be made consistent for all cities. If this is not possible add a discussion of uncertainties due to the a-priori settings. Error bars only account for statistical day-to-day variations, but ignore the impact of a-priori settings and the CER procedure.

*Author Response:*

*Thank you for your time and suggestions, particularly concerning details of the method that we used, which helped to improve the manuscript.*

**Detailed comments:**
**- Line 117: Is this bias in NO2 accounted for in your study? How would the results change if NO2 would be scaled up accordingly?**

*Author Response:*

*The bias in S5P NO2 retrieval has been assessed for European cities. However, since we don't know yet how representative this estimate is for the cities that we study, it was decided to account for the impact of the bias as an additional the source of uncertainty of 30% of the TROPOMI inferred NO2/CO ratio (see Table S3).*

*If the 30 % bias would apply to all the cities, then the TROPOMI derived emission ratios increase by factor of 1.30. In that case, TROPOMI-derived ratios remain in line with the emission inventories but the agreement would shift between EDGAR and MACCity depending on the city, for example in favor of EDGAR in case of Cairo and Riyadh. For Los Angeles, the difference between TROPOMI and the inventories would decrease, but remain significantly lower for TROPOMI.*

**Table 1 settings: I am puzzled by the different definitions for different cities, especially as Line 146 states that the settings are "not critical". So why are they different at all? Do you need to tune the area definitions in order to get the right results??? How do the results look like if a consistent setup is chosen for all cities? Why is the upwind area for Riyadh different for dlat and dlon by a factor of 30?**

*Author Response:*

*The following sentence was added (line 154-163): "Every city has a different size and different neighboring CO and NO2 emission sources and therefore the appropriate choice of radii for the background and outskirt areas varies between cities (see Supplements Section 1 for details). Since the same regional definition is used for NO2 and CO, the enhancement ratio is not sensitive to the details of the region selection. Most important for the choice of radii is to catch the local enhancement in CO and NO2 to its full extend, to optimize the signal over noise and thereby the detection limit for urban emissions."*

*We performed a sensitivity test in which the background and outskirt radii were increased by 10km in four steps resulting in the ratio changes by < 15 % for all the cities (see Figure S20).*

*There was a typing error for Δlat and Δlon of Riyadh. To maintain the consistency now, I am using 1.0˚, 1.0˚ Δlat and Δlon for all the cities.*

**Page 6:**
the methods are explained in plain words, but not illustrated for real data. So Fig. S2 should be moved to the main text, and the background/upwind regions etc. should be marked in this plot. In addition, the ERA wind vector should be added. The rotated patterns and the percentiles used for the second approach should be provided in a separate figure.
*Author Response:*
*Figure1 is added in the paper, to illustrate the upwind background method for real data as suggested by the reviewer. For the plume rotation method, see the Figure S8.*

[Figure]

**Figure 1.** Average wind speed and direction from the surface to 200m from ERA Interim at the time of TROPOMI overpasses (left) and TROPOMI derived total column CO over Mexico City (right) for June 4$^{th}$, 2018. The black star represents the centre of the city. In the right panel, the white circle is the background area for Mexico City and the blue section represents the upwind background area that we selected depending upon the wind direction ($\theta$) in the core city area. P0, P1, P2 and P3 are the points where the north, east, west and south wind directions intersect with the inner rim of the background area. P0$_{new}$ is the new point generated by rotating P0 with $\theta$ in reference to the city centre

[Figure]

**Figure S8.** TROPOMI derived XCO a) averaged over June-August 2018 b) plume rotated over Mexico, c) Upwind and d) downwind region. The white star in top panel is the centre of Mexico City. The white lines in panel (b) represent the $20\times100\text{km}^2$ area to determine the column enhancement in the city. The area to the North and South of the city centre is upwind and downwind region respectively. The red line in panel c) and d) represents the 25 percentile and 75 percentile respectively. CO retrievals are gridded in $0.1°\times0.1°$.

**both methods compare columns "upwind" and "downwind" of the investigated cities. This approach requires that there *is* transport taking place. Wind speeds for Mexico City are quite low, as can alsobe seen in Fig. S2. So did you consider a minimum threshold for the wind speed? I expect that it would help to remove inconclusive days.**

*Author Response:*
*In this study we do not use a minimum threshold for the wind speed, because depending on the choice which would inevitably be quite arbitrary and a substantial amount of useful data may be lost. However, as explained before, by studying the ratio between NO2 and CO we are less sensitive to transport issues. Most important is to have a method the quantifies the difference between city and background in a way that is consistent between CO and NO2.*

**- Line 157: with Eqs 2&3, daily ratios are calculated. But how is the total ratio (shown in Fig. 3) derived? Is it the mean of all daily ratios? This is by definition different from the second approach, where first CO and NO2 are averaged and then the ratio of means is calculated. Thus, also for approach 1, the ratio of means should be taken.**
*Author Response:*

*In the upwind background method, the mean is taken of daily ratios. This method is favorable over the use of mean NO2 and CO in the plume rotation method, because it accounts for temporal correlations between NO2 and CO. However, for the plume rotation method we choose to stay consistent with Pommier et al (2013), which averages first since it is important when using noisy MOPITT data. The implication is that there is the inconsistency that the reviewer mentions. However, by including it in the comparison, we also implicitly test the robustness of the TROPOMI derived ratio to this methodological difference.*

**- Line 196: What emission database is used by CAMS? EDGAR? MACCity ? Or something else ? How far does this affect the following interpretation and discussion of CAMS OH? How do CAMS spatial patterns of CO and NO2 compare to TROPOMI? Please provide a Figure in the Supplement. Is TROPOMI CO and/or NO2 assimilated in CAMS?**
*Author Response:*
*CAMS uses MACCity for the anthropogenic emission. CAMS OH depends upon the chemical scheme used for the simulation in addition to the emission inventories (V. Huijen et al., 2019). We do not discuss or interpret CAMS OH. We only use it to derive an estimate of the NO2 lifetime, which is valid within the uncertainty bounds of the reanalysis. The spatial patterns of CAMS and TROPOMI NO2 are in reasonably good agreement for the six different cities (see Section 5 in Supplements Fig S12 to Fig S17). However, larger differences are found for CO over Tehran, Cairo, Riyadh and Lahore (see Section 5 in Supplements Fig S14 to Fig S17). CAMS is using MOPPIT for CO and for NO$_2$, SCIAMACHY, GOME2 and OMI data are assimilated. TROPOMI CO and NO2 are not yet included in CAMS.*

[Figure]

**Figure S12.** Collocated XCO (top) and XNO2 (bottom) averaged for June-August, 2018 over Mexico City comparing TROPOMI (left) and CAMS (right). The white star represents the centre of the city.

[Figure]

**Figure S13.** Same as FigS12 but over Los Angeles

[Figure]

**Figure S14.** Same as Fig S12 for Tehran.

[Figure]

**Figure S15.** Same as Figure S12 but for Cairo

[Figure]

**Figure S16.** Same as Figure S12 but for Riyadh

[Figure]

**Figure S17.** Same as Figure S12 but for Lahore

**2.6.1: The bootstrapping approach evaluates the statistical uncertainty for the results with the chosen approach. But on top, there are also other uncertainties, like systematic effects introduced by the definition of radii etc. In particular the uncertainties of the wind direction and wind speed have to be discussed as well.**

*Author Response:*
*As suggested by the reviewer, we tested the sensitivity to the wind speed and direction by choosing different heights (i.e. 200m to 1000m), resulting in differences ≤10 % for all the cities (see Figure S18, S19).*

[Figure]

**Figure S18.** TROPOMI derived Upwind Background Corrected Emission Ratio for six megacities using average wind speed and direction calculated from surface till 200m to 1000m. The error bar represents 1σ uncertainties calculated using boot strapping.

[Figure]

**Figure S19.** Same as Fig S11 but the emission ratio is Plume Rotation Corrected Emission ratio

*We also performed a sensitivity analysis in which the background and outskirt radii were increased by 10km in four steps resulting in an uncertainty of 7 to 15 % in TROPOMI column enhancement ratios for the six megacities (see Figure S20).*

[Figure]

**Figure S20.** Upwind Background Corrected Emission ratio derived for six megacities using four different background and outskirt radius. For the initial step the outskirt and background radius for Tehran: 180 km and 190 km, Mexico City: 170km and 180km, Cairo: 135km and 145 km, Riyadh: 100km and 110 km, Lahore: 100km and 110km and Los Angeles: 200 km and 210 km. In every step the background and outskirt radius is increased by 10km.During this process dlat and dlon is 1.0˚,1.0 ˚.

**Figure 3: Please estimate the uncertainties of NO2 lifetime and AK correction and provide error bars for the CER results as well. I expect that these uncertainties are far higher (and thus more relevant) than the purely statistical bootstrap uncertainties.**

*Author Response:*
*Table S3 shows the different sources of uncertainty (i.e. NO2 lifetime, Ak correction, wind direction and their contribution to the TROPOMI derived emission ratio). Error bars indicating 1σ uncertainties in TROPOMI derived emission ratios are added in Figure 4. The following sentence is added to line 355 to 360 explaining the different sources of uncertainty and their effect on TROPOMI derived emission ratios:*

*"Additionally, TROPOMI underestimates the $NO_2$ column by 7 % to 29.7 % relative to MAX-DOAS ground based measurement in European cities (Lambert, et al., 2019). However, since we don't know yet how representative this estimate is for the cities that we study, the impact of this bias has been accounted for as an additional source of uncertainty of 30% in the TROPOMI inferred NO2/CO ratio (see Table S3). Compared to this number, other sources of uncertainty such as in the wind direction and speed (FigS18 and S19), boundary layer OH concentration, $A_{influence}$ correction and the predefined background setting (Fig S20) make only small contributions to the TROPOMI derived emission ratio. The total uncertainty in the TROPOMI derived emission ratio is calculated using error propagation (see Table S3) and ranges between 33 to 35.6%."*

**Table S3. Estimated uncertainties in TROPOMI derived emission ratios. The total uncertainty is derived by summing the individual components in quadrature**

| City | S5P TROPOMI $NO_2$ uncertainty (A) (%) | Wind direction and Wind speed (B) (%) | Boundary layer OH concentration (C) (%) | Predefined background area (D) (%) | $A_{influence}$ Correction (E) (%) | Total effect on ER $(\sqrt{(A^2 + B^2 + C^2 + D^2 + E^2)}$ (%) |
|---|---|---|---|---|---|---|
| Tehran | 30 | 10.5 | 8.4 | 3.0 | 1.2 | ±33.03 |
| Mexico City | 30 | 7.5 | 10 | 9.0 | 2.7 | ±33.83 |
| Cairo | 30 | 2.3 | 8.4 | 13.0 | 4.4 | ±34.12 |
| Riyadh | 30 | 6.0 | 12.5 | 9.1 | 4.1 | ±34.5 |
| Lahore | 30 | 6.5 | 15.0 | 10.0 | 0.4 | ±35.6 |
| Los Angeles | 30 | 10.0 | 8.3 | 8.8 | 4.2 | ±34.1 |

**- 3.2ff: Please check the discussion and conclusions (a) for NO2 probably being biased low and (b) according to the quality of CAMS emissions and the agreement between TROPOMI and CAMS**
*Author Response:*

*Discussion section: Lines 355-360 discuss the importance of the bias in NO2. The sentence is as follows:" Additionally, TROPOMI underestimates the NO$_2$ column by 7 % to 29.7 % relative to MAX-DOAS ground based measurement in European cities (Lambert, et al., 2019). However, since we don't know yet how representative this estimate is for the cities that we study, the impact of this bias has been accounted for as an additional source of uncertainty of 30% in the TROPOMI inferred NO2/CO ratio (see Table S3)."*

*CAMS use multiple datasets i.e. MACCity emission inventory, SCIAMACHY, GOME2 and OMI data for CO and NO2. We don't know the influence of each of these datasets on the result. Therefore, we decided not to include CAMS in the comparison to TROPOMI derived emission ratios.*

*Conclusion Section:*
*- Lines 435 to 436 explain about the uncertainties due to NO2 being biased low:" The total uncertainty on TROPOMI derived emission ratio ranges from 33 to 35.6%.The bias in S5P TROPOMI NO2 retrievals has the most important contribution to the uncertainty in the TROPOMI derived emission ratio".*

**Minor comments:**
**- Lines 40-42: The references stating the high uncertainty of Chinese emissions are from a time period where development in China was vastly increasing. Meanwhile, NOx emissions have been reduced, and the awareness of air pollution has increased in China. I would thus assume that these high uncertainties do not hold any longer.**
*Author Response:*

*- We have modified the sentence to clarify which years the uncertainty estimate referred to.*

**- Lines 60-61 Please provide refs to SCIAMACHY (Bovensmann) and TROPOMI (Veefkind).**
*- Changed as suggested*

**- Line 70: Should be NOx emission.**
*- Changed as suggested*

**- Line 75: Transport disperses NO2 and CO similarly, but the lifetime of NO2 is far shorter! See the different plume extents shown in Fig. S2.**
*Author Response:*
*The reviewer is right that the plume extents are different due to the difference on lifetime, which is the reason why we focus on the core city area rather than the full extent of the plume.*

**- Line 117: Avoid misreading as "the bias is low", e.g. "NO2 is biased low by about**

*- Changed as suggested*

**- Table 1 lat/lon: Please provide consistent number of digits for lat/lon. .01_ should be accurate enough.**
*- changed as suggested*

**- Fig. S1: I don't understand why there is a need for separating 4 different wind directions in the formalism; rotation matrix should work the same for all four cases!?**

*Author Response:*

*-changed as suggested and see Figure S7.*

[Figure]

**Figure S7.** Schematic representation of the procedure used for selecting the upwind background. The centre of the city is represented by red star. The city, outskirt and background radii (see Table 1) are used to divide the city into three parts i.e. the core city (red circle), outskirt (white circle) and background region (blue circle), respectively. Step 1.Selection of radius R1, the mean of outskirt and background radii. Select the points P0, P1, P2, P3 where the north, east, south and west wind directions ($\theta$) intersect at the outer rim of the dashed circle with radius R1. The black arrow symbolises an average wind direction over the core city region. Step2. The rotation of P0 with $\theta$ in reference to the city centre and generate the new point (P0$_{new}$).Step 3. Select the square box in the background area using $\Delta$lat and $\Delta$lon provided in Table1. Step 4. Rotate the square box using wind direction ($\theta$) in the core city area in reference to the P0$_{new}$ and select the fraction of the upwind region.

---

## Author Comment (AC7) · 3 Jul 2020

Please note that Figures are in supplement attached below.

This study presents new results from TROPOMI for NO2/CO emission factors that provide information about combustion efficiency on urban scales. This is an important result for understanding how well these emissions are represented in standard inventories with subsequent impacts for air quality and climate model predictions. I recommend publication after the comments from 2 other referees and some minor issues from me are addressed.

[Figure]

Author Response:

Thank you for your time and comments to improve our paper.

1. Following the comment of Ref.#1 in addressing the different NO2 and CO lifetimes, the different seasonality in concentrations should also be addressed. For example, is seasonality removed before computing the background CO? Also, in computing emission inventory ratios, are monthly emissions used when matching to data from a particular month, or do you apply annual averages?

Author Response:

To address this point, we switched to monthly emissions using EDGAR v4.3.2 2010 and MACCity 2018. The following text was added to line 344– 350:"Seasonal variations in emission factors may influence our comparison between seasonal averaged TROPOMI data and annual average EDGAR emissions. To account for the influence of seasonally varying emission factors, we compute a seasonal correction factor based on EDGAR v4.3.2 2010 since monthly data are not available for EDGAR 2012(see Fig.4). Except for Lahore, the June to August (JJA) EDGAR ratio is lower by 5 to 12.5 % compared to the annual average EDGAR ratio. The MACCity ratio for JJA, however, is higher by 10 to 71% compared to the annual average, indicating that EDGAR and MACCity disagree on the seasonality of the NO2/CO emission ratio. For MACCity, the agreement with TROPOMI improves the most when taking seasonality into account (see Fig.4)."

2. Abstract. The abstract should state that NO2/CO is a proxy for combustion efficiency since combustion efficiency is a well-defined quantity: CO2/(CO2+CO). This would be better than calling it "burning efficiency", which is confusing since combustion and burning are the same.

Author Response:

In the introduction section Line 74 to 77: "We use the ratio of the TROPOMI retrieved tropospheric column of NO2 and the total column of CO, which is formally not equivalent to combustion efficiency but can nevertheless serve as a useful proxy of the burning conditions (Silva and Arellano, 2017; Tang and Arellano, 2017). The reason for this is that the NOx emission increases with combustion temperature, which is high during efficient combustion. In contrast, CO is a product of incomplete combustion, and is produced when combustion efficiency is low (Flagan & Seinfeld, 1988).The combination of these effects makes the NO2/CO ratio highly sensitive to combustion efficiency" make the things clear about the combustion and burning efficiency.

Furthermore, abstract lines 15 to 17 have been changed into: " Efficient combustion is characterized by high NOx (NO+NO2) and low CO emissions, making the NO2/CO ratio a useful proxy for combustion efficiency."

3. Perhaps the title could be: "Quantifying NO2/CO using TROPOMI to characterize urban combustion"

Author Response:

Thank you for the suggestion however, we deliberately do not use the term 'combustion efficiency'. Instead, we choose to keep the old formulation of the title and explain carefully in the introduction section what we mean by burning efficiency

4. Line 57 – should also reference Tang et al., 2019:

-changed as suggested

5. Line 85 – MOPITT also has a SWIR channel (or near IR) and the multispectral (TIR/NIR) product, with near-surface sensitivity over some land regions, was used in both Silva and Arellano, 2017 and Tang and Arellano, 2017.

-changed as suggested

Please also note the supplement to this comment:
https://www.atmos-chem-phys-discuss.net/acp-2019-1112/acp-2019-1112-AC7-supplement.pdf

[Figure]

**Supplement:**

This study presents new results from TROPOMI for NO2/CO emission factors that provide information about combustion efficiency on urban scales. This is an important result for understanding how well these emissions are represented in standard inventories with subsequent impacts for air quality and climate model predictions. I recommend publication after the comments from 2 other referees and some minor issues from me are addressed.

*Author Response:*

*Thank you for your time and comments to improve our paper.*

Following the comment of Ref.#1 in addressing the different NO2 and CO lifetimes, the different seasonality in concentrations should also be addressed. For example, is seasonality removed before computing the background CO? Also, in computing emission inventory ratios, are monthly emissions used when matching to data from a particular month, or do you apply annual averages?

*Author Response:*

*To address this point, we switched to monthly emissions using EDGAR v4.3.2 2010 and MACCity 2018. The following text was added to line 344– 350:"Seasonal variations in emission factors may influence our comparison between seasonal averaged TROPOMI data and annual average EDGAR emissions. To account for the influence of seasonally varying emission factors, we compute a seasonal correction factor based on EDGAR v4.3.2 2010 since monthly data are not available for EDGAR 2012(see Fig.4). Except for Lahore, the June to August (JJA) EDGAR ratio is lower by 5 to 12.5 % compared to the annual average EDGAR ratio. The MACCity ratio for JJA, however, is higher by 10 to 71% compared to the annual average, indicating that EDGAR and MACCity disagree on the seasonality of the NO2/CO emission ratio. For MACCity, the agreement with TROPOMI improves the most when taking seasonality into account (see Fig.4)."*

[Figure]

**Figure 4.** Comparison of TROPOMI-derived ΔNO₂/ΔCO enhancement ratios, calculated using different methods shown in blue shades, to corresponding emission ratios from the EDGAR (red shades) and MACCity (yellow shades) emission inventories for six megacities. Dark solid shades for emission inventories represent the annual average inventory derived ratio, whereas faded shades represent June to August averaged inventory derived ratios. The upwind background corrected emission ratio (UBCER) and Plume rotation corrected emission ratio (PRCER) account for the impact of photochemical NO₂ removal and the averaging kernel. Error bars for TROPOMI-derived ΔNO₂/ΔCO enhancement ratios represent 1σ uncertainties calculated using boot strapping (upwind background) and error propagation (plume rotation method) .The error bar for UBCER and PCER accounts the uncertainty in methodology and TROPOMI data (for details see Table S3).

**Abstract. The abstract should state that NO2/CO is a proxy for combustion efficiency since combustion efficiency is a well-defined quantity: CO2/(CO2+CO). This would be better than calling it "burning efficiency", which is confusing since combustion and burning are the same.**

*Author Response:*

*In the introduction section Line 74 to 77: "We use the ratio of the TROPOMI retrieved tropospheric column of NO₂ and the total column of CO, which is formally not equivalent to combustion efficiency but can nevertheless serve as a useful proxy of the burning conditions (Silva and Arellano, 2017; Tang and Arellano, 2017). The reason for this is that the NOₓ emission increases with combustion temperature, which is high during efficient combustion. In contrast, CO is a product of incomplete combustion, and is produced when combustion efficiency is low (Flagan & Seinfeld, 1988).The combination of these effects makes the NO₂/CO ratio highly sensitive to combustion efficiency" make the things clear about the combustion and burning efficiency.*

*Furthermore, abstract lines 15 to 17 have been changed into: " Efficient combustion is characterized by high NOx (NO+NO$_2$) and low CO emissions, making the NO$_2$/CO ratio a useful proxy for combustion efficiency."*

**Perhaps the title could be: "Quantifying NO2/CO using TROPOMI to characterize urban combustion"**

*Author Response:*

*Thank you for the suggestion however, we deliberately do not use the term 'combustion efficiency'. Instead, we choose to keep the old formulation of the title and explain carefully in the introduction section what we mean by burning efficiency*

**Line 57 – should also reference Tang et al., 2019:**

*-changed as suggested*

**Line 85 – MOPITT also has a SWIR channel (or near IR) and the multispectral (TIR/NIR) product, with near-surface sensitivity over some land regions, was used in both Silva and Arellano, 2017 and Tang and Arellano, 2017.**

*-changed as suggested*